# Moisture transport and Antarctic sea ice: Austral spring 2016 event

Monica Ionita[1], Patrick Scholz[1], Klaus Grosfeld[1], Renate Treffeisen[1]

[1]Alfred Wegener Institute Helmholtz Centre for Polar and Marine Research, Bremerhaven, 27570, Germany

*Correspondence to*: Monica Ionita (Monica.ionita@awi.de)

**Abstract.** In austral spring 2016 the Antarctic region experienced anomalous sea ice retreat in all sectors, with sea ice extent in October and November 2016 being the lowest in the Southern Hemisphere over the observational period (1979 – present). The extreme sea ice retreat was accompanied by widespread warming along the coastal areas as well as in the interior of the Antarctic continent. This exceptional event occurred along with a strong negative phase of the Southern Annular Mode (SAM) and the moistest and warmest spring on record, over large areas covering the Indian Ocean, the Ross Sea and the Weddell Sea. In October 2016, the positive anomalies of the totally integrated water vapor (IWV) and 2m air temperature (T2m) over the Indian Ocean, Western Pacific, Bellingshausen Sea and southern part of Ross Sea were unprecedented in the last 39 years. In October and November 2016, when the largest magnitude of negative daily sea ice concentration anomalies were observed, repeated episodes of poleward advection of warm and moist air took place. These results suggest the importance of moist and warm air intrusions into the Antarctic region as one of the main contributors to this exceptional sea ice retreat event.

## 1 Introduction

Sea ice in both polar regions is an important indicator for the expression of global climate change and its polar amplification. It also plays an important role in modulating the global climate system by influencing the atmospheric and oceanic circulations (Cohen et al., 2018). Moreover, it has also a strong impact on the global economic system through changes in marine and natural resources development. Consequently, a broad scientific as well as societal interest exists on sea ice, its coverage, variability, and long term change (National Research Council, 2012). Recent observed changes in the Arctic have become the "face" for global climatic changes, especially due to the rapidly decreasing trend of the summer sea ice extent over the last two decades (Serreze and Stroeve, 2015). The trends in the Arctic sea ice extent (SIE) over the satellite observational record 1979 – present are negative for all months, with the largest trend recorded at the end of the melt season in September (Serreze et al., 2007; Stroeve et al., 2015), with an average decline of 12.9% per decade relative to the long-term mean of 1981 – 2010 September average (Grosfeld et al., 2016).

In contrast, over the Antarctic region the SIE shows an increasing trend over the satellite observational record, seemingly at odds with climate model projections. The increase in the Antarctic sea ice extent can be largely explained by natural climate fluctuations (Meehl et al., 2016), linked with different factors/drivers like: atmospheric temperature or wind stress (Liu et al., 2004; Turner et al., 2009; Lefebvre and Goosse, 2005), precipitation (Liu and Curry, 2010), ocean temperature (Jacobs and

Comiso, 1997), and feedbacks in the atmosphere or ocean (Stammerjohn et al., 2012; Zhang, 2007). The variability of the Antarctic SIE has been also related to several large-scale atmospheric circulation mechanism like: a) the southern annular mode (SAM) (Hall and Visbeck, 2002; Kwok and Comiso, 2002; Lefebvre et al., 2004; Simpkins et al., 2012; Raphael and Hobbs, 2014); b) the zonal wave 3 (ZW3) (Raphael, 2007, Schlosser et al., 2018) and c) the El Niño –Southern oscillation

(Turner, 2004; Yuan, 2004; Stammerjohn, 2008).

Whereas the coupled atmosphere-ocean-sea ice models, which are driven by realistic natural and anthropogenic forcing, are able to replicate the decreasing Arctic trend (Stroeve et al., 2012), they also have the tendency to simulate a significant decrease in Antarctic sea ice cover (Turner et al., 2013), in contrast with the observed increasing trend. Shu et al. (2015) showed that only one out of seven CMIP5 models is able to simulate the sign of the Antarctic sea ice trend correctly. As the

confidence on numerical simulations relies on an adequate representation of the late-twentieth-century changes (Collins et al., 2014), the understanding of the physical processes generating the Antarctic SIE trend and variability is crucial for the relevance of the projected Southern Hemisphere sea ice extent decrease. Consequently, the analyses of observed, reconstructed, and reanalysis data are critical for the investigation of their causes. Following this line, here we analyze the exceptional Antarctic austral spring (September – October – November) of 2016, from a climatological point of view and try

to identify the drivers responsible for triggering this particular event. This study is an additional contribution in the discussion about the possible drivers of the 2016 extremely low sea ice event over the Antarctic region. Turner et al. (2017) considered the spatial differences of the SIE anomalies and their temporal change at regional level. They related the low SIE over the Antarctic region to the warm air advection and the strong negative values of SAM in November and December 2016. Schlosser et al. (2018) have shown that the rapid decrease in the sea ice area and extent were associated with

atmospheric flow patterns reminiscent to a positive zonal wave number 3 (ZW3) index. The strong meridional flow associated with a positive ZW3 index triggered an accelerated sea ice decline, especially at the beginning of November. From a pre-conditioning point of view, Stuecker et al. (2017) showed that the extreme low SIE in November and December 2016 was partially driven by the 2015/16 El-Niño event and the negative phase of SAM.

Due to the fact that the Arctic and Antarctic regions are moisture flux convergence areas, atmospheric moisture transport is a

primary input source of water into these regions. Moreover, through the cloud-radiative forcing the moisture transport directly or indirectly affects the snow, sea ice and ice sheet over the polar regions. For the Arctic region, Kapsch et al. (2013) demonstrated that a significantly enhanced transport of humid air during spring produces increased cloudiness and humidity, thus accelerating the sea ice retreat in summer. Therefore, atmospheric moisture transport is a crucial component for the water balance, especially over the polar regions. While for the Arctic regions there is a large number of studies dealing with

the influence of the moisture transport on the sea ice variability (Mortin et al., 2016; Yang and Magnusdottir, 2017; Park et al., 20125), over the Antarctic region little attention has been paid to transport variations of moisture from the Extratropics (Nieto et al., 2017). As such, the objectives of our paper are as follows: (a) to characterize the temporal and spatial extent of the spring 2016 exceptional sea ice melting event using both daily and monthly sea ice data; (b) to analyze the key drivers of the event, with a special emphasis on the role played by enhanced moisture transport and warm intrusions into the Antarctic

region; (c) to place the austral spring 2016 into a long-term perspective. The paper is structured as follows: in Section 2, we introduce the data used in this study; the main results of our analysis are shown in Section 3, while the concluding remarks are presented in Section 4.

## 2 Data and methods

### 2.1 Data

For the daily sea ice concentration (SIC) we utilize the passive microwave sea ice concentration data using the National Snow and Ice Data Center (NSIDC) bootstrap algorithm (Comiso and Nishio, 2008; Meier et al., 2013) over the period 1979 – 2016. Prior to 1987, data are only available every other day; hence data for missing days were produced by linear interpolation for each pixel between the fields of the previous and next days. The linear interpolation is a standard procedure

of this sea ice product. The monthly SIE index (Fetterer et al., 2016) over the Antarctic region has been extracted from the NSIDC ftp server ([ftp://sidads.colorado.edu/DATASETS/NOAA/G02135/north/](ftp://sidads.colorado.edu/DATASETS/NOAA/G02135/north/)). Sea ice extent (SIE) is defined as the total area of all satellite pixels where the sea ice concentration equals or exceeds 15%. Following previous studies (Zwally et al., 2002; Turner et al., 2017), we examine the anomalies of the daily and monthly SIC and SIE for the Southern Ocean as a whole and for five separate sectors (Figure 1): the Ross Sea (RS) (160°E–130°W), Amundsen - Bellingshausen Sea (ABS)

(130°W–60°W), Weddell Sea (WS) (60°W–20°E), Indian Ocean (IO) (20°E–90°E), and Western Pacific Ocean (WPO) (90°E–160°E).

For the Southern Hemisphere (SH) temperature and atmospheric circulation, we use the monthly means of air temperature at 2m (T2m), zonal wind at 10 m (U), meridional wind at 10 m (V), mean sea level pressure (SLP), the vertical integral of water vapour (IWV), the vertical integral of eastward water vapour flux ($Q_u$), the vertical integral of northward water vapor

flux ($Q_u$), the vertical integral of eastward heat flux ($H_u$), and the vertical integral of northward heat flux ($H_v$) from the European Centre for Medium-range Weather Forecasts (ECMWF) Interim (ERA-Interim) reanalysis fields (Dee et al., 2011). IWV, $Q_u$, $Q_v$, $H_u$ and $H_v$ are produced by the ERA post processing budget software. Details how the basic fields are post-processed are given in the IFS documentation ([https://www.ecmwf.int/sites/default/files/elibrary/2017/17738-part-vi-technical-and-computational-procedures.pdf](https://www.ecmwf.int/sites/default/files/elibrary/2017/17738-part-vi-technical-and-computational-procedures.pdf)). ERA-Interim uses a four-dimensional variational assimilation scheme with a

twelve-hour analysis window. The data assimilation system is based on a 2006 version of the ECMWF Integrated Forecast Model. The spatial resolution of the model is T255 (~80 km) on 60 vertical levels from the surface up to 0.1 hPa (Dee et al., 2011). The data are continuously updated in real time from 1st January 1979 to present. The ERA-Interim fields are reliable across high southern latitudes from 1979 and are considered to be the best reanalysis data set for depicting recent Antarctic climate (Bracegirdle and Marshall, 2012). They are typically considered to have the highest quality relative to other

reanalysis data from the water cycle, with a better representation of mass fluxes including water vapour (Lorenz and

Kunstamnn, 2012). ERA-Interim dataset has been also shown to significantly improve humidity fields in Antarctica and the Southern Ocean (Uppala et al. 2008; Simmons et al. 2010).

The vertically integrated water vapor transport (WVT) (Peixoto and Oort, 1992) is defined as follows:

$$\vec{Q}(u, v, t) = Q_u \vec{i} + Q_v \vec{j} \tag{1}$$

where $Q_u$ is the vertical integral of eastward water vapour flux and $Q_v$ the vertical integral of northward water vapour flux.

The station based monthly mean temperature data has been extracted from the Reference Antarctic Data for Environmental Research (READER). The primary sources of data are the Antarctic research stations and automatic weather stations. The stations used in the current study have been downloaded from the following website: https://legacy.bas.ac.uk/met/READER/data.html.

For the Southern Annular Mode (SAM) index, we used the index from https://legacy.bas.ac.uk/met/gjma/sam.html. SAM index is available over the period 1957 up to date and the index is computed from surface meteorological observations from Antarctic coastal and Southern Ocean island stations (Marshall, 2003). SAM is the dominant mode of low-frequency atmospheric variability in the Southern Hemisphere. The SAM index measures a "see-saw" of atmospheric mass between the middle and high latitudes of the Southern Hemisphere. Positive values of the SAM index correspond with stronger-than-

average westerlies over the mid-high latitudes (50S-70S) and weaker westerlies in the mid-latitudes (30S-50S) (Marshall, 2003). All the anomalies (SIC, SLP, TT, IWV, QU, QV, HU and HV) used in this study were computed relative to the climatological period 1981–2010.

**2.2 Methods**

The *stability maps* approach used in this study is based on a methodology that was successfully applied to predict the monthly to seasonal streamflow of Elbe and Rhine rivers (Ionita et al., 2008, 2015, 2017; Meißner et al., 2017). The basic idea of the stability maps is to identify regions with stable teleconnections (the correlation does not change in time) between SIE averaged over specific regions and meteorological/climatological gridded data. As such, we correlate the regional sea ice index with IWV and T2m gridded fields in a 21-years moving window. The correlation is considered to be *stable* for those

grid points where the sea ice index and the gridded fields are significantly correlated at 95%, 90%, 85%, and 80% significance level, for more than 80% of the 21-years moving windows. The regions where the correlation is positive and stable will be represented as dark red (95% significance level), red (90% significance level), orange (85% significance level) and yellow (80% significance level) on the stability maps (see Figure 6 and 7). The regions where the correlation between the regional SIE index and the gridded data is stable and negative will be represented as dark blue (95% significance level),

blue (90% significance level), green (85% significance level) and light green (80% significance level) on the stability maps (see Figure 6 and 7). For the current analysis only regions, where the correlation is above 95% significance level, are retained for further analysis. The results remain qualitatively the same if the length of the moving window varies between 15 and 25 years. Although the length of our time series is relatively short (40 years) the methodology proved to work also in cases of timer series < 40 years (Ionita et al., 2017a). Moreover, we use the same methodology, with the same number of

years (40 years), for the prediction of September Arctic and Antarctic Sea Ice (https://www.arcus.org/sipn/sea-ice-outlook/2017/post-season). A more detailed description of the *stability maps* approach can be found in Ionita et al. (2008) and Ionita (2017a).

## 3 Results

### 3.1 Daily and monthly Antarctic sea ice extent

Antarctic sea ice has shown increased SIE, area, and concentration from the late 1970's until 2015. Even though the trend itself is modest (Yuan et al., 2017), it is somewhat challenging to explain in the context of the overall global warming signal. When looking at the SIE anomalies over the last 38 years, the first eight months of 2016 are not particularly anomalous (not shown). Both July (Figure S1a) and August (Figure S1b) 2016 were characterized by slightly positive pan-Antarctic SIE anomalies. However, starting from September 2016, the situation changed dramatically. The Antarctic SIE from September 2016 until December 2016 was characterized by significant negative anomalies, with November 2016 and December 2016 ranking as the lowest SIE for those respective months in the sea ice record (Figure S1). November mean SIE was more than five standard deviation below the 1981-2010 average (Stammerjohn and Scambos, 2017). September 2016 ranks as the 7[th] in term of lowest SIE on record, while October 2016 ranks as the second in terms of lowest SIE.

To have a clear picture of the spatial pattern of the extreme events in the austral spring 2016, we have computed the SIC anomalies in each grid point, from September 2016 until December 2016 (Figure 1). In September 2016, most of the Antarctic region was characterized by negative SIC anomalies (Figure 1a), with some exceptions over the eastern part of the WS and the eastern part of RS, where positive SIC anomalies prevailed. The same spatial pattern was present also in October 2016 (Figure 1b), but with much higher amplitudes (SIC anomalies < 40%) over the IO, WS, and the western part of RS. In November 2016, the negative SIC anomalies become very high in amplitude over the western part of the WS and western part of ABS (Figure 1c). Weak positive SIC anomalies were present over the western part of WS, as well as over the north-eastern part of RS, and south-eastern part of ABS. Over the WPO, the SIC anomalies reversed their sign (negative to positive) in November compared to October 2016. In December 2016, the whole RS was characterized by negative SIC anomalies (Figure 1d), as well as ABS (with some small exceptions over the eastern part) and IO.

When looking at the daily evolution of the SIE anomalies throughout the austral spring 2016, some striking features are present. In September 2016, most of the negative SIE anomalies at the circumpolar Antarctic level were mainly driven by the daily SIE anomalies over IO and ABS (Figure S2). In October 2016, all regions, except Weddell Sea, were characterized by daily negative SIE anomalies, especially IO. The situation became very dramatic in November 2016, when daily negative SIE anomalies were recorded over all the analyzed regions (Figure S2), especially IO and WS (in the second part of November). The combined effect of all regions, showing daily negative SIE anomalies throughout entire November 2016, was a SIE anomaly below -2.2 Mio. km$^2$ around 20[th] of November. In December 2016, the extreme daily SIE anomalies continued the decreasing trend, with a SIE anomaly below -2.7 Mio. km$^2$ occurring in the middle of the month. Most of the

contribution for the extreme SIE anomalies in December 2016 was coming from the WS. By analyzing the daily SIE anomalies throughout the year 2016, one of the most striking features is the abrupt drop of SIE of more than 1.7 Mio. km$^2$ from the beginning of November 2016 until the middle of December 2016, which is unprecedented since the beginning of the 1980's (not shown).

## 3.2 Climatological analysis of the austral spring 2016

In September 2016, the atmospheric circulation features a center of negative SLP anomalies over the Antarctic continent flanked by three centers of positive SLP anomalies: one over the RS, one over IO, and one over WS. This SLP pattern was associated with positive temperature anomalies over the ABS, WS, IO, and eastern WS and negative temperature anomalies

10 over most of the Antarctic continent, except the Antarctic Peninsula (Figure 2a). The three positive SLP centers were positioned in such a way that they enhanced the poleward advection of moist (Figure 3a) and warm air (Figure 4a), especially over IO, eastern RS, WS, and Bellingshausen Sea (Figure 2e). The areas, characterized by positive temperature anomalies and enhanced water vapor, correspond to the regions where the negative SIC anomalies, were occurring in September 2016 (see Figure 1a).

15 In October 2016, the atmospheric circulation was more meridional and wavier, with altering positive and negative SLP anomalies surrounding the Antarctic continent (Figure 2b). SAM had a value of -0.89, ranking as the 14$^{th}$ lowest SAM index on record. This wavy SLP pattern favored, again, the advection of moist (Figure 3b) and warm air (Figure 4b) towards the continent and the coastal areas, the most affected regions in terms of sea ice reduction, corresponding to the regions where warm and moist intrusions occurred: WS, IO, ABS, and western WPO. October 2016 was also characterized by widespread

20 warm anomalies along the coastal areas as well as in the interior of the continent, with some exceptions (Figure 2f). In November 2016, SAM index dropped to a value of -3.12, which ranks as the fifth lowest SAM index since 1958 (Stamerjohn, 2016). In terms of large-scale atmospheric circulation, an anomalous high pressure system developed over the polar cap, which resulted in the weakening of the westerlies. The high pressure system over the southern polar cap was surrounded by a band of negative SLP anomalies, stretching from IO - WPO - RS up to ABS (Figure 2c). Extreme positive

25 temperature anomalies were recorded over the whole southern polar cap and the coastal areas (Figure 2h). Moreover, over ABS, WS, and RS, enhanced advection of moist (Figure 3c) and warm (Figure 4c) air from the lower latitudes led to a rapid melting of the sea ice over these regions (Figure 1c).

In December 2016, the atmospheric circulation was similar to November 2016: high pressure system over the southern polar cap, surrounded by negative SLP anomalies over WS and IO (Figure 2d). The December value of the SAM index (-1.52,

30 10th lowest for December since 1957) was much smaller compared to November 2016. This is also visible in the amplitude of the SLP anomalies, over the polar cap, between the two months, with December 2016 featuring weaker (in amplitude) positive SLP anomalies over the Antarctic Continent and smaller temperature anomalies (by a factor of 4 between November 2016 and December 2016). In December 2016, positive (but weak) temperature anomalies prevailed over most of the

Antarctic Continent, except for the central part (Figure 2h). Moist (Figure 3d) and warm (Figure 4d) conditions were also recorded over RS and parts of the coastal regions of IO and WPO (Figure 2h).

**3.3 Long-term context of the austral spring 2016 event**

To demonstrate that austral spring 2016 was a very exceptional year in terms of reduced SIC and warm and moist conditions over the Antarctic region and the surrounding areas, we have computed the rank maps for the monthly (September – December) SIC (Figure 5 – first column), IWV (Figure 5 – middle column), and T2m (Figure 5 – third column) over the period 1979 -2016. The red areas in Figure 4 indicate that 2016 was the year with the lowest SIC, the moistest, and the warmest year on record, over the period 1979 – 2016. The orange colors indicate that 2016 was the second lowest year in SIC, moisture, and warmth and so on. For clarity we have chosen only the top 5 ranks. The idea of ranking maps has been successfully used before to emphasize the "exceptionality" of a particular year/season/month (Ionita et al., 2017).

September 2016 was the month with the lowest SIC over distinct areas in the WS, RS, and ABS, and ranks among the top five years with the lowest SIC over an extended band in the central IO (Figure 5a). The regions with the lowest SIC, in September 2016, were also the moistest (Figure 5b) and warmest (Figure 5c) on record. The regions with the lowest SIC in October 2016 (Figure 5d) are similar to the ones from September 2016, but the spatial extent of low SIC anomalies is larger compared to September 2016. October 2016 stands out as the moistest and warmest year on record over large areas covering the surroundings of Antarctica: almost the entire IO region stands out as the moistest and warmest on record, with small exceptions in the western part of it; the southern part of RS was also among the moistest and warmest regions on record, while the whole Bellingshausen Sea stands out in the first five moistest and warmest October on record (Figure 5e and 5f). The largest regions affected by low SIC were recorded in November 2016 (Figure 5g), with central IO, south-eastern RS, and small parts of WS being the areas with the lowest SIC on record. In terms of moist and warm, the regions that stand out as very extreme, in November 2016, are located over the western part of RS, and the eastern part of Antarctica as well as over the WS (Figure 5h and 5i). Around the coastal areas of East Antarctica all stations where measurements are available recorded the highest monthly mean temperatures in October and November 2016 (Casey: -3.7°C; Davis: -2.2°C, Syowa: -4.7°C, Figure 6a and 6b) (Keller et al., 2017). In December 2016, the spatial extent of the extreme low SIC is much smaller (Figure 5j) compared to the previous months, being restricted just to the southern part of RS and eastern part of ABS. The same is valid for IWV and TT. The north-eastern part of RS ranks as the moistest on record (Figure 5k), while small regions in WS rank as the warmest on record (Figure 5l).

**3.4 Long term relationship between regional Antarctic sea ice, moisture availability and temperature**

As shown in Section 3.3, the period October - December 2016 was characterized by large positive anomalies in the surface temperature and the vertical integral of water vapor covering large areas in the Antarctic region (Figure 2 and Figure 3). To examine the long-term relationship between moisture and temperature on one hand and the regional Antarctic sea ice variability on the other hand we have computed the stability maps between IO SIE (November) and IWV and T2m (October

and November), as well as the stability maps between WS SIE (December) and November and December IWV and T2m (November and December), over the period 1979 – 2016. We performed the analysis, both in phase as well as with 1-month lag (IWV and T2m leading the regional SIE index) to have a clear picture of the relationship between SIE and IWV and T2m. We opted for November IO SIE and December WS SIE due to the fact that the highest SIE anomalies were recorded

over these particular regions in 2016 (Figure S2 and Table 1).

The stability maps between November IO SIE and October/November IWV and T2m are shown in Figures 7a-d. In general, negative SIE anomalies over IO, in November, tend to occur in association with enhanced moisture and positive temperature anomalies over IO region in the previous month. In agreement with the findings in Section 3.3, year 2016 stands out as the most extreme one both in terms of SIE and moisture availability. Based on the stable regions identified in Figures 7a-d

(marked by yellow boxes) we defined two indices: one for October IWV (IWV averaged over the yellow box in Figure 7a) and one for October T2m (T2m averaged over the yellow box in Figure 7c). For the in-phase stability maps (Figures 7b and 7d), the stable regions are very small and very regional and we did not consider them for further analysis. The correlation coefficient between November IO SIE and October IWV index is r = -0.62 (99% significance level), while the correlation coefficient between November IO SIE index and October T2m index is r = -0.63 (99% significance level).

The stability maps between December WS SIE and November/December IWV and T2m are shown in Figure 7a-d, respectively. In the case of December WS SIE, the stability maps show extended stable regions, having a dipole-like structure, both for November IWV and November T2m (marked by yellow and green boxes). Positive SIE anomalies over WS, in December, are associated with negative IWV and T2m anomalies over the WS region and positive IWV and T2m anomalies over RS, in the previous November. Based on the stability maps in Figure 8a-d, we have defined five indices

(yellow and green boxes in Figure 8a, 8c and 8d). As in the case of November IO, the stable regions are more extended for 1-month lag analysis. The correlation coefficient between December WS SIE and November IWV1 index is r = -0.68 (99% significance level) and the correlation between December WS SIE index and November T2m1 index is r= -0.60 (99% significance level). The correlation coefficient between December WS SIE and November IWV2 (T2m2) index is r = 0.34 (0.45), while the correlation coefficient between December WS SIE and December T2m index is r = -0.35. Negative SIE

anomalies, over WS in December, are occurring in association with enhanced moisture and positive temperature anomalies over the WS and reduced moisture and negative temperature anomalies over RS, in the previous month.

The significant lagged relationship between the regional SIE and previous month IWV and T2m can be used as potential predictive information for the development of the upcoming SIE over particular regions. The same analysis can be performed for other regions or at pan-Antarctic level and by including other climate/oceanic variables. Since the focus of our

paper was to see the influence of the moisture and temperature on the development of the Antarctic sea ice extreme event in the austral spring of 2016, here we focused just on IWV and T2m.

## 3.5 Daily evolution of the integrated water vapor transport in 2016

So far we have considered only monthly means, but a signal that is visible at monthly time scale has to have been predominant over a part of the month or extreme over particular days throughout the month (Schlosser et al., 2018). The later was the case for the moisture intrusions towards the Antarctic region, in October and November 2016, respectively. To emphasize the importance of these extreme daily moisture intrusions, based on the rank maps shown in Figure 4, we have defined two indices over the regions which rank as the moistest on record in October and November (black squares in Figure 5e and 5f). These indices have been computed by spatially averaging the daily vertical integral of water vapor over the months of October and November from 1979 until 2016. The two regions are defined over IO (Figure 9a) and WS (Figure 10a).

Over IO, there were two extreme events occurring in October 2016: one on the 3$^{rd}$ of October (Figure 9b) and the second one occurring on the 26$^{th}$ of October (Figure 9c). For these two extreme events the vertical integral of water vapor shows a band of enhanced moisture stretching from the IO towards the coastal eastern part of Antarctica (Figure 9b and 9c). The regions with the highest IWV are located over the IO (black squares in Figure 9b and 9c). Both events show clearly narrow bands, where the moisture fluxes are concentrated. These kind of events, also known as atmospheric rivers (ARs), have been associated with the occurrence of anomalous snow accumulation in east Antarctica (Gorodetskaya et al., 2014). In general, ARs transport large amounts of warm and humid air.

In November 2016, there were also two extreme events occurring (in terms of moisture transport): one on the 3$^{rd}$ of November (Figure 10a and 10b) and the second one on the 16$^{th}$ of November (Figure 10a and 10c). Although the IWT index is defined over a restricted area in the WS, when looking at the vertically integrated total moisture transport (vectors in Figure 9b and 9c) and the associated IWV values, a very particular picture emerges. During these two particular events, the WS, IO, WPO and ABS were under the influence of the advection of moist and warm air from the mid-latitudes. WS received enhanced moisture from the Tropical South Atlantic Ocean, IO received a band of enhanced moisture both from the Tropical South Atlantic Ocean as well as from the IO, and WPO received a lot of moisture from the IO. The combined effect of this enhanced moisture advection from the tropics, which happened simultaneously in WS, IO, WPO and ABS could, at least partly, explain the extremely low values of the daily SIE throughout the whole month of November 2016 and the beginning of December (Figure S2). Looking at the daily vertically integrated total moisture transport, we can observe that moisture sources are coming from the tropical parts of the Pacific Ocean (for ABS and RS), Atlantic Ocean (for WS and IO), and Indian Ocean (for IO and WPO). This is in agreement with a recent study of Drumond et al. (2016), who show that the most important moisture sources for the Antarctic ice cores are: the subtropical South Atlantic Ocean, Indian Ocean, and South Pacific Ocean.

## 4 Discussion and conclusions

An initial attribution of the Antarctic conditions throughout the 2016 austral spring indicates that the atmospheric circulation in the SH was very anomalous. The SAM was negative from October 2016 until December 2016, with top negative value of

-3.12 in November 2016 (the fifth lowest SAM since 1958). The persistence of the negative phase of SAM throughout these three months is associated with a weakening of the circumpolar westerlies and an overall surface warming of the coastal parts of the Antarctic continent, as well as large parts inside the continent. The overall warming accompanied by with enhanced poleward advection of warm and moist air led to strong sea ice melting, especially in October and November.

Tietäväinen and Vihma (2008) have shown that poleward water vapor transport, especially over the RS, WS, and IO, is usually associated with the SAM wave number three. This is in good agreement with our findings, especially for the months of October and November 2016.

The results presented in our study can be regarded as an additional contribution regarding the drivers of the 2016 extremely low sea ice event over the Antarctic region. In previous studies regarding the 2016 event, Turner et al. (2017) related this low

SIE to the warm air advection and the strong negative values of SAM in November and December. Schlosser et al. (2018) have shown that the rapid decrease in the sea ice area and extent were associated with atmospheric flow patterns reminiscent to a positive zonal wave number 3 (ZW3) index, which triggered accelerated sea ice decline, especially at the beginning of November. Stuecker et al. (2017) showed that the extreme low SIE in November-December 2016 was partially driven by the 2015/16 El-Niño event and the negative phase of SAM. In our study, we show that enhanced moisture and positive

temperature anomalies over different parts of the Antarctic region (e.g., IO, WS, RS) have led to the extremely low SIE recorded throughout the austral spring 2016. In October and November 2016, when the lowest daily SIC anomalies were observed, repeated episodes of poleward advection of warm and moist air took place, with the most intense episodes occurring in October and November 2016. Moreover, October 2016 ranks as the moistest and warmest October over the last 39 years, over large areas in the IO, ABS and RS (Figure 5e and 5f), while November ranks as the moistest and warmest

November over extended parts over WS, WP and WS (Figure 5h and 5i). In this study, we have shown that negative SIE anomalies at regional level (e.g. IO and WS) are preceded by enhanced moisture and positive temperature anomalies over these regions, one month ahead. The lagged relationship between the regional SIE anomalies and previous month IWV and T2m could be used to provide a potential predictive skill for the upcoming development of the regional sea ice extent anomalies.

Given the dramatic SAM anomalies in some months, we investigated also the long term relationships between regional SIE and the monthly SAM (Table 2). However, significant relationships were found only for three month-region combinations and thus consistent with previous studies (Stammerjohn et al., 2008; Lefebre et al., 2004; Holland et al., 2017; Liu et al., 2004; Simpkins et al., 2012). In September, significant correlations between SAM and regional SIE are found just for WS ($r = -0.37$, 95% significance level), while for the other regions, the correlations are not significant. Throughout the month of

October, no significant correlations are found between monthly SAM index and the regional Antarctic SIE, while for November significant correlations are found just between the SAM index and IO ($r = 0.43$, 99% significance level). In December, significant correlations are found between the SAM index and RS ($r = 0.46$, 99% significance level). Overall, positive SAM in September occurs in association with negative SIE anomalies over WS, while in November positive SAM tend to be associated with positive SIE over IO. In December, a positive SAM index is associated with positive SIE over RS.

Our analysis indicates that SAM's role in sea ice variability is rather complex and dependent of the region under consideration.

Although at Antarctic level no significant trends in the meridional moisture fluxes over the period 1979-2010 have been observed (Tsukernik et al., 2013), the austral spring 2016 stands out as very extreme in this respect, with 2016 ranking as the moistest (Figure S3a) and warmest austral spring on record (Figure S3b), over large areas covering Antarctica and the surrounding seas. The same holds for the Arctic region, boreal autumn 2016 ranking as the moistest and warmest autumn on record over almost the whole Arctic basin (Figure S3c and Figure S3d). Generally speaking, both austral spring and boreal autumn 2016 have been characterized by enhanced poleward advection of moist and warm air. This exceptional event, occurring simultaneously in both polar regions, might be a direct or indirect consequence of the combined effect of 2015/16 El Niño event (Stuecker et al., 2017) and the fact that 2016, as a whole, was the warmest year on record (NOAA, 2017). Together, these two exceptional events could have triggered the release of large amounts of moisture in the atmosphere and via particular large-scale atmospheric circulation patterns (e.g. Rossby waves, weaker jet stream) parts of this moisture have been carried out towards the poles.

In order to have a clear picture of the physical mechanisms which relates IWV and T2m to the variability of the Antarctic SIE complementary modelling simulations are needed. By using observational/reanalysis data alone it is difficult to disentangle which of the potential drivers (e.g., IWV and T2m) has a stronger impact on the sea ice variability. For the Arctic region, previous studies (Mortin et al., 2016; Kapsch et al., 2013; Park et al., 2015) suggested that the transport of heat and moisture into the Arctic during spring enhances the incoming surface longwave radiation, thereby controlling the initiation of the annual ice melt and setting the stage for the Arctic September ice minimum. Thus, our study can be used as an indicator that moisture transport should be considered in future studies as a potential driver of the Antarctic sea ice variability.

From the perspective of future climate projection, our results imply that a realistic simulation of the Antarctic SIE trend and variability requires also a proper simulation of both climate modes of variability (e.g., SAM) as well as extreme events (e.g., moisture intrusions, Rossby waves, atmospheric blocking). Concluding, the 2016 austral spring event demonstrates that the present-day climate of the Antarctic continent and the surrounding areas allow for extensive sea ice reduction to occur. If such events are going to occur more frequently in the future is uncertain, but it is essential to understand and highlight the mechanisms responsible for triggering such events.

*Acknowledgements*. This study is promoted by Helmholtz funding through the Polar Regions and Coasts in the Changing Earth System (PACES) program of the AWI. Funding by the Helmholtz Climate Initiative REKLIM is gratefully acknowledged. P. Scholz has been found by he Collaborative Research Centre TRR 181 "Energy Transfer in Atmosphere and Ocean" funded by the German Research Foundation.

30

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

30

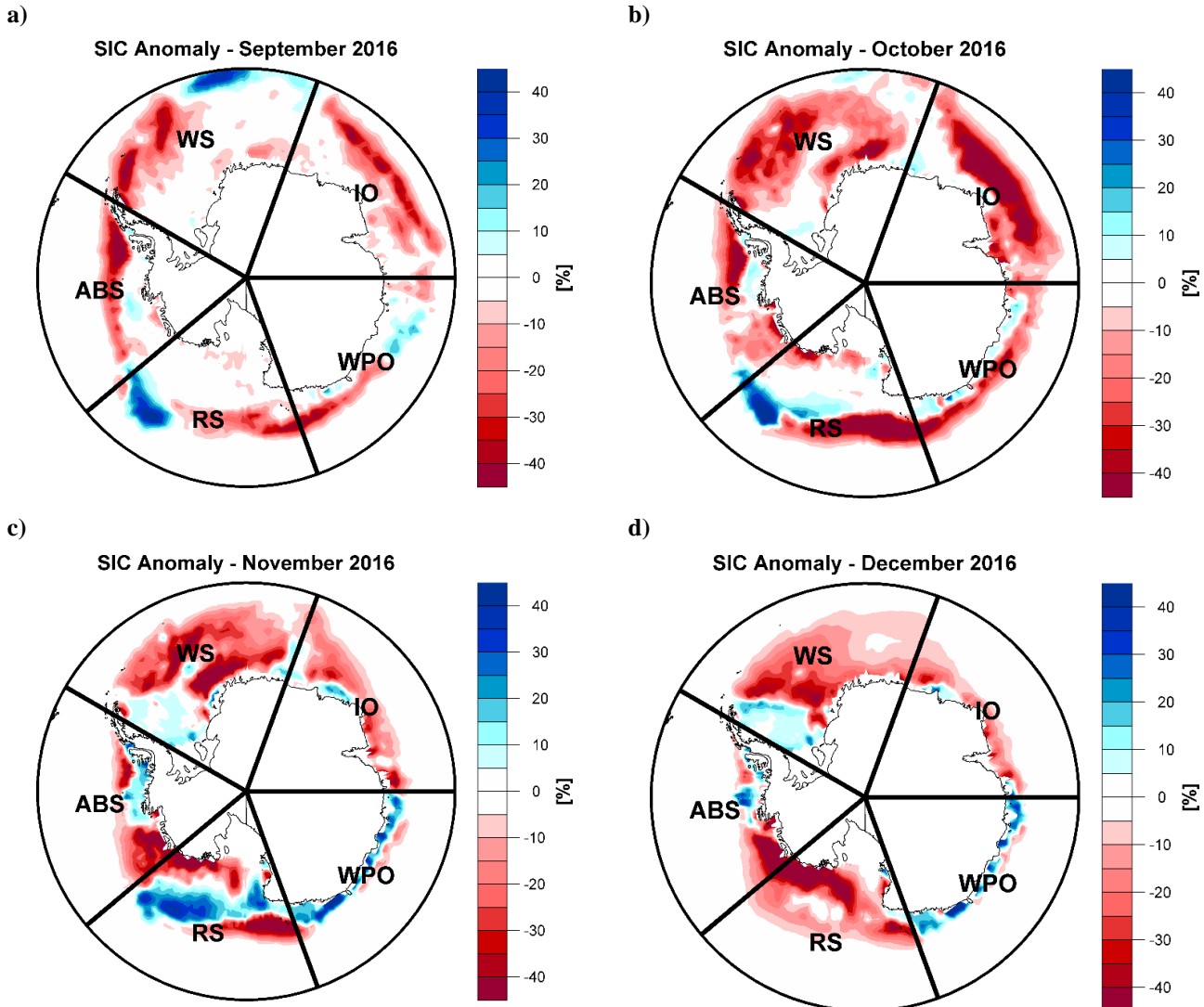

**Figure 1.** Sea ice concentration (SIC) anomalies 2016 for: a) September; b) October; c) November and d) December. The anomalies are computed relative to the reference period 1979 – 2010. WS = Weddell Sea; IO = Indian Ocean; WPO = Western Pacific Ocean; RS = Ross Sea; ABS = Amundsen - Bellingshausen Sea.

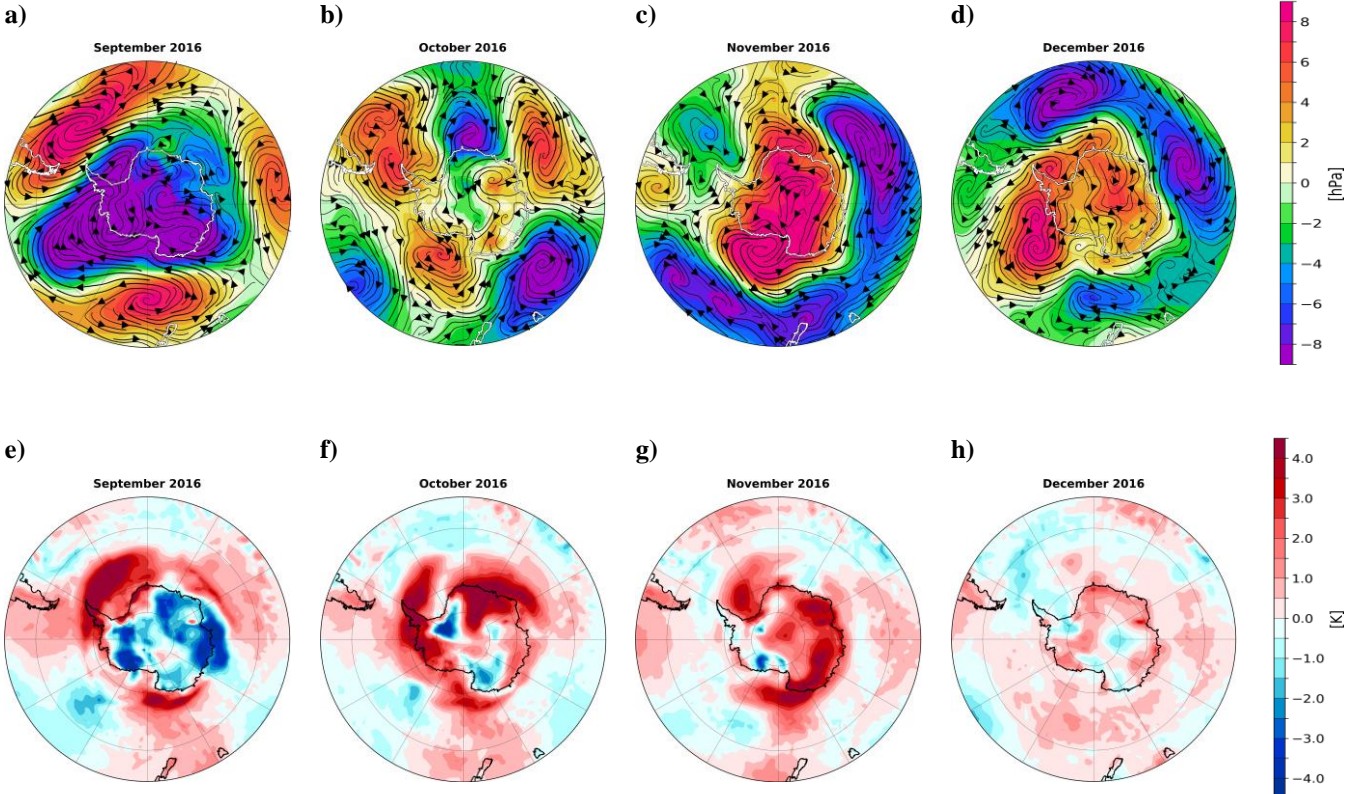

**Figure 2.** Sea level pressure (SLP) anomalies and the associated wind streamlines (upper panels) and 2m air temperature (T2m) anomalies (lower panels) from September until December 2016. The anomalies are computed relative to the reference period 1979 – 2010.

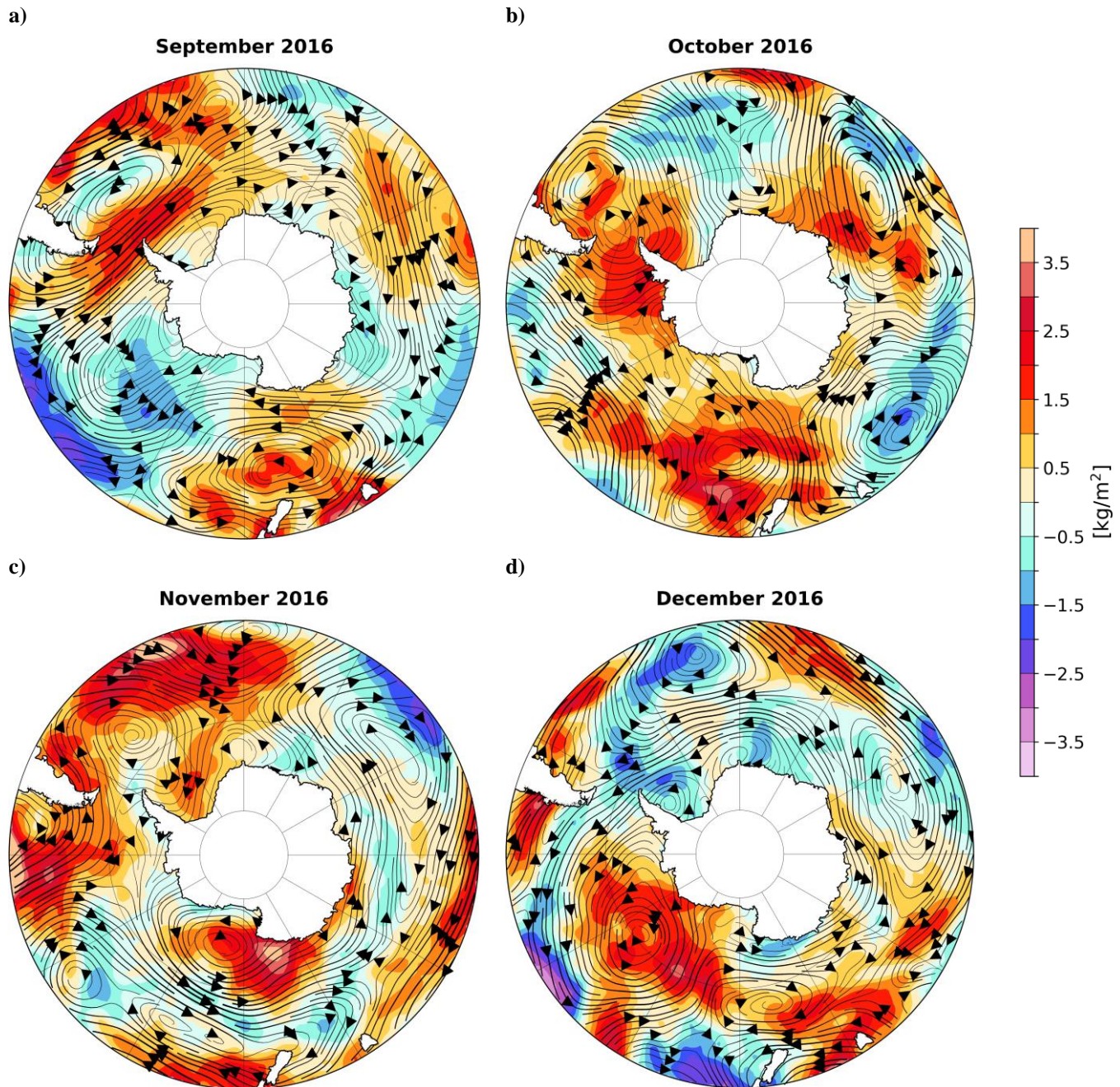

**Figure 3.** The vertical integral of water vapor anomalies (IWV, shaded areas) and the vertically integrated water vapor transport (WVT, streamlines) from September until December 2016. The anomalies are computed relative to the reference period 1979 – 2010.

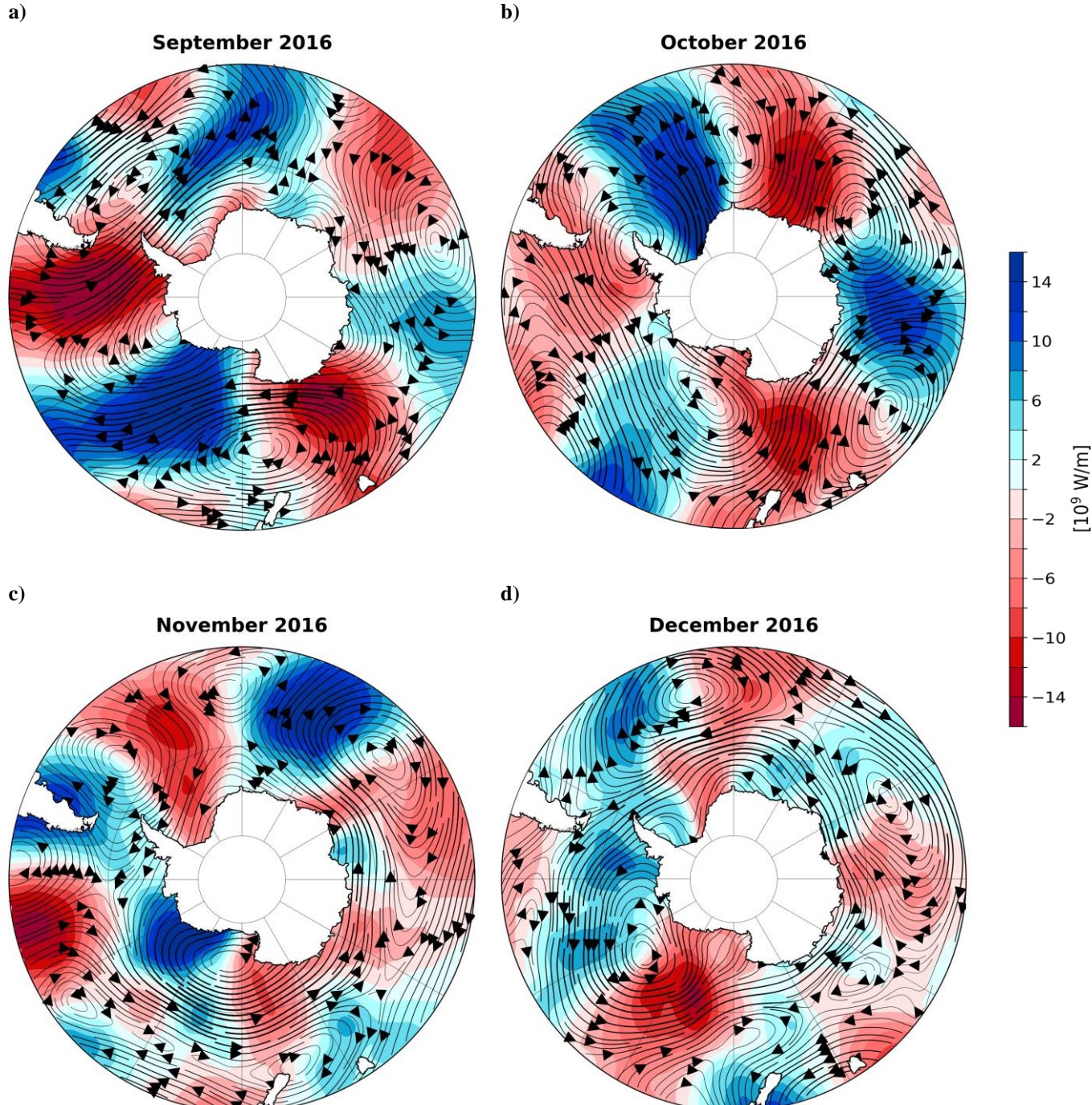

**Figure 4.** The vertical integral of northward heat flux (shaded areas) and the vertically integrated heat transport (streamlines) (lower panels) from September until December 2016. The anomalies are computed relative to the reference period 1979 – 2010. In a) – d) positive anomalies indicate northward advection and negative indicates southward advection.

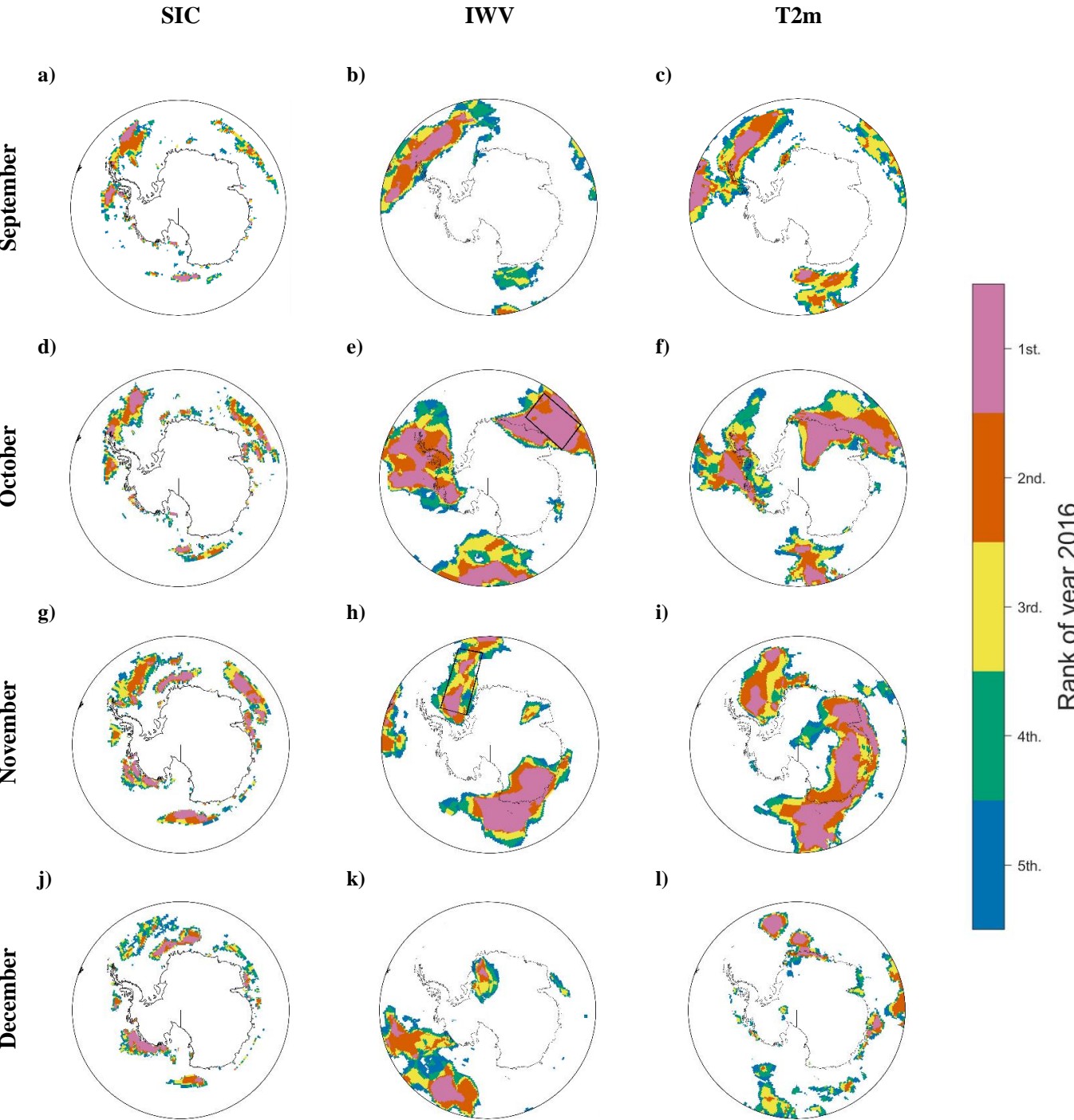

**Figure 5.** Ranking of 2016 monthly: Sea ice concentration (SIC - first column, '1' means lowest SIC over the analyzed period), the vertical integral of water vapor (IWV – second column, '1' means the moistest month over the analyzed period); 2m air temperature (T2m – third column, '1' means the warmest month over the analyzed period). Analyzed period: 1979–2016. Rankings below 5 appear white.

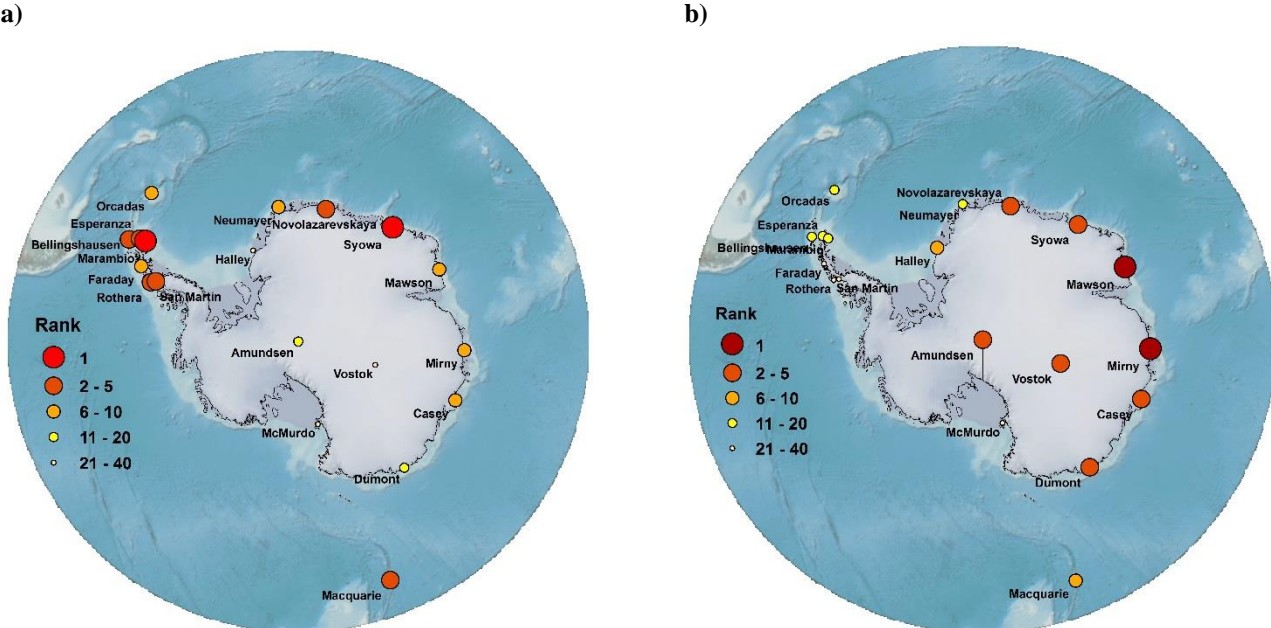

**Figure 6.** Ranking of the monthly mean temperature at observation stations: a) October 2016 and b) November 2016. Number "1" means the warmest month since 1981, number "2" signifies the second warmest, etc. Analyzed period: 1981–2016.

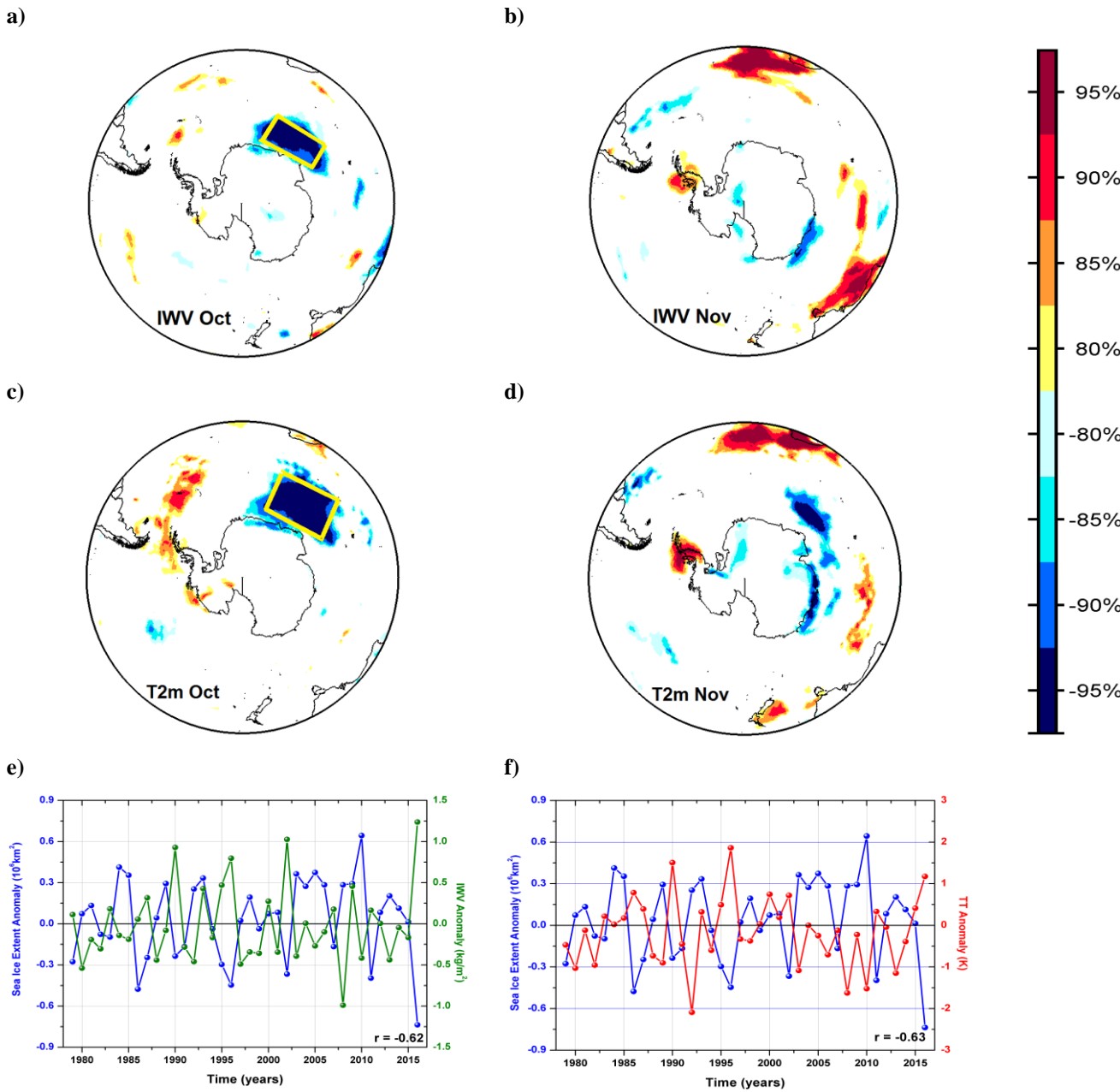

**Figure 7.** The stability maps between November Indian Ocean (IO) sea ice extent and a) October IWV; b) November IWV; c) October T2m; d) November T2m; e) The time series of November IO SIE (blue line) and the time series of October IWV (green line) averaged over the yellow box in a); f) The time series of November IO SIE (blue line) and the time series of October T2m (red line) averaged over the yellow box in c). The regions where the correlation is positive and stable are represented as dark red (95% significance level), red (90% significance level), orange (85% significance level) and yellow (80% significance level) and the regions where the correlation between the regional SIE index and the gridded data is stable and negative will be represented as dark blue (95% significance level), blue (90% significance level), green (85% significance level) and light green (80% significance level) on the stability maps.

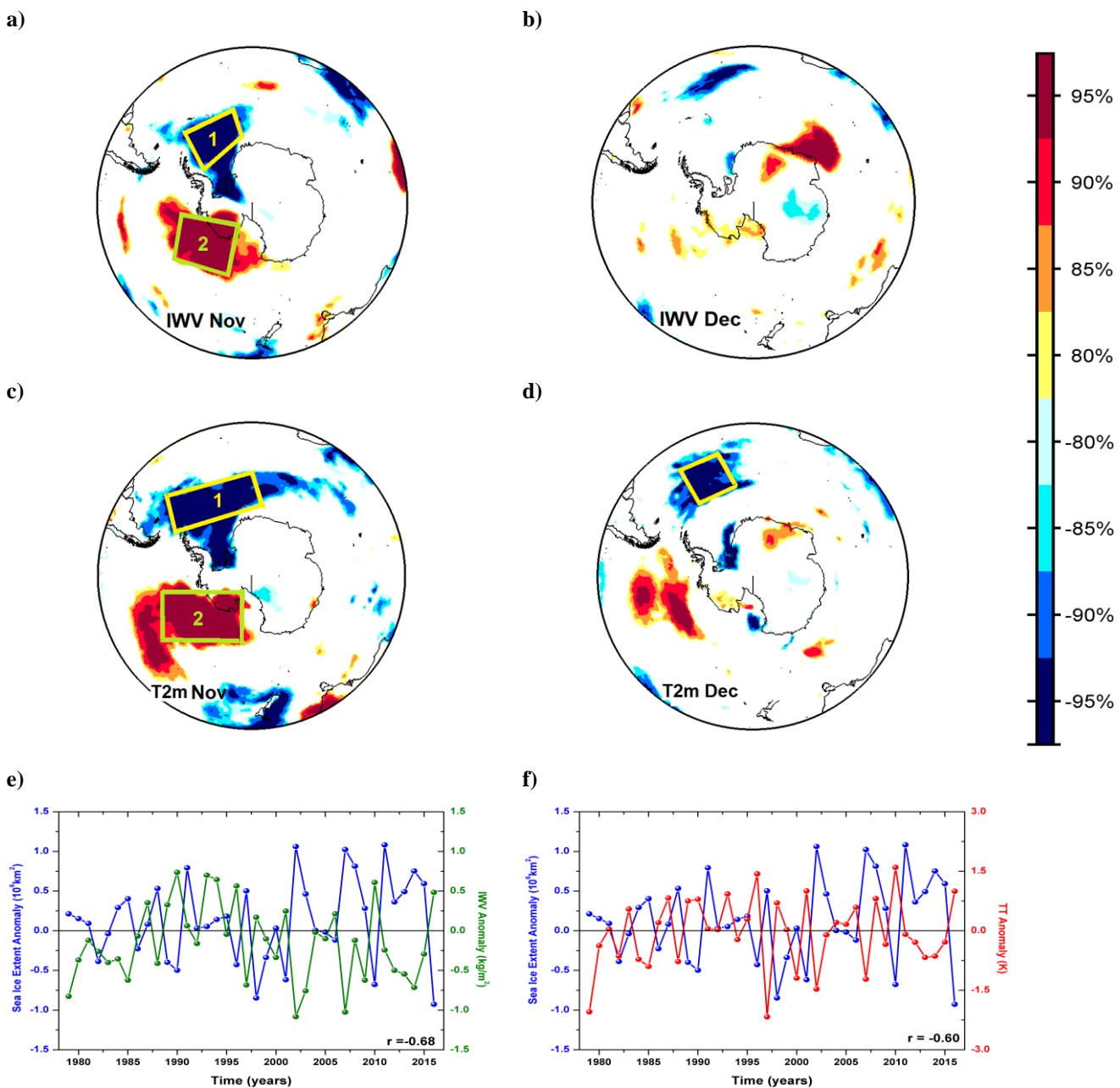

**Figure 8.** The stability maps between December Weddell Sea (WS) sea ice extent and a) November IWV; b) November T2m; d) December T2m; e) The time series of December WS SIE (blue line) and the time series of November IWV1 (green line) averaged over the yellow box in a); f) The time series of December WS SIE (blue line) and the time series of November T2m1 (red line) averaged over the yellow box in b). The regions where the correlation is positive and stable are represented as dark red (95% significance level), red (90% significance level), orange (85% significance level) and yellow (80% significance level) and the regions where the correlation between the regional SIE index and the gridded data is stable and negative will be represented as dark blue (95% significance level), blue (90% significance level), green (85% significance level) and light green (80% significance level) on the stability maps.

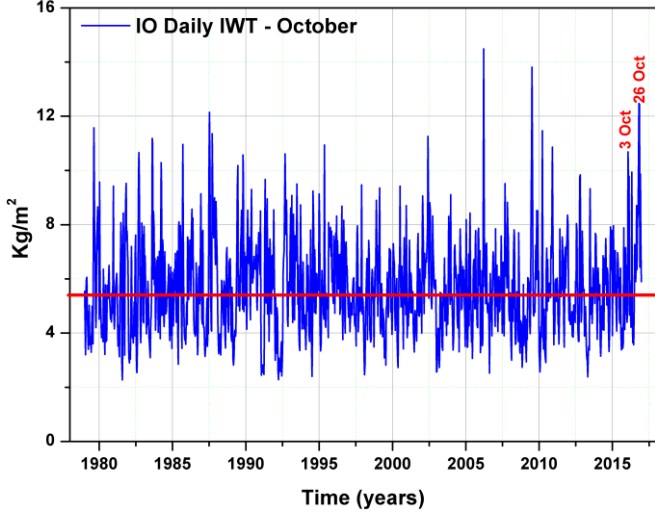

                                                                          

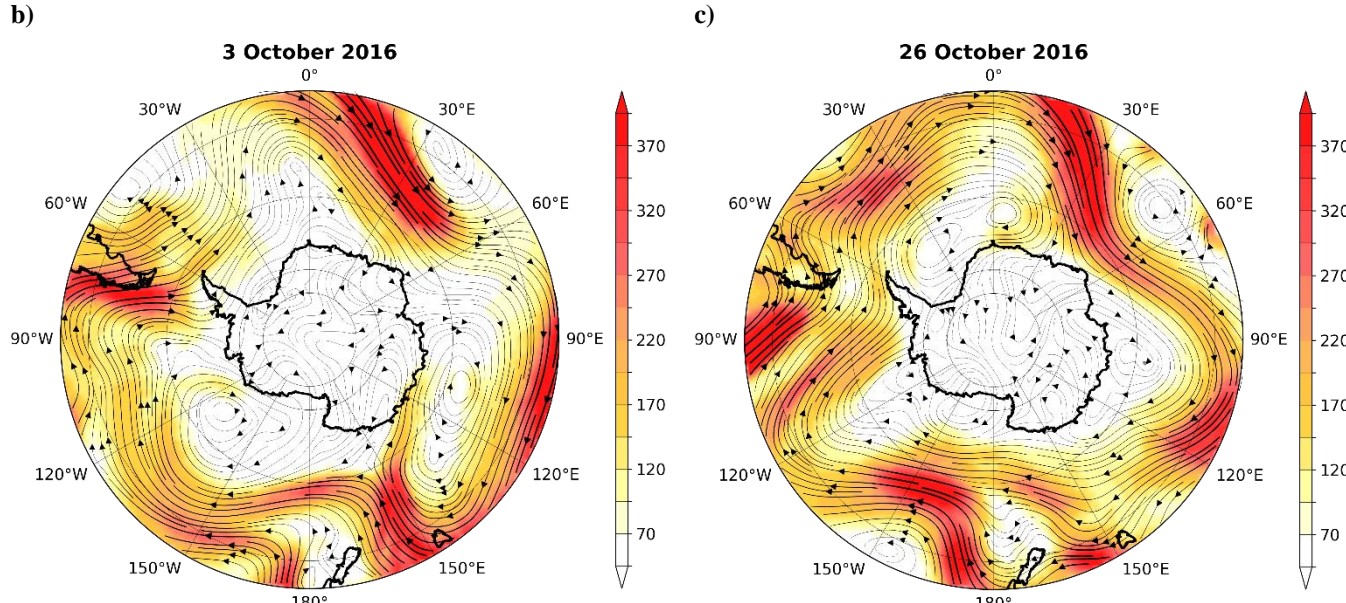

**Figure 9.** a) Daily IWT averaged over IO (black square in Figure 4e) over the period 1979 – 2016; b) The vertically integrated water vapor transport (WVT) on the 3rd October 2016 and c) as in b) but for the 26th October 2016. In b) and c) the color shaded areas indicated the magnitude of the moisture transport and the vectors the direction of the moisture transport.

a)

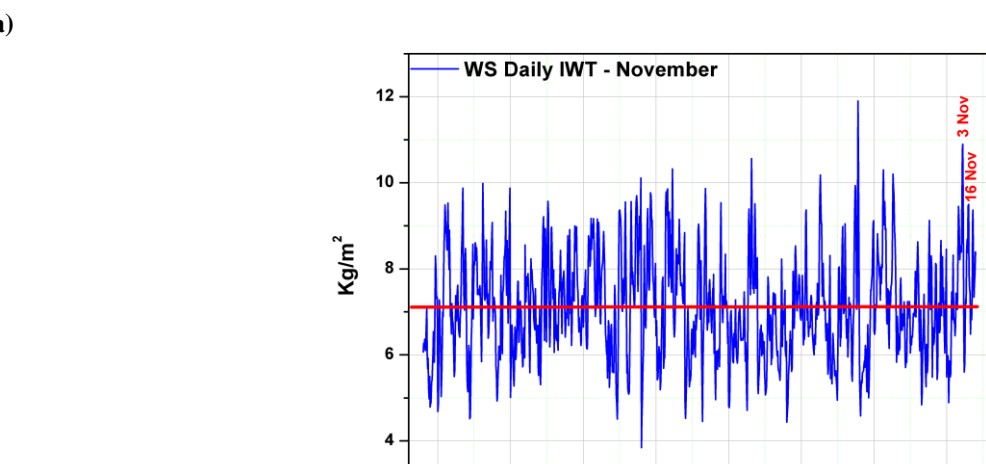

b) c)

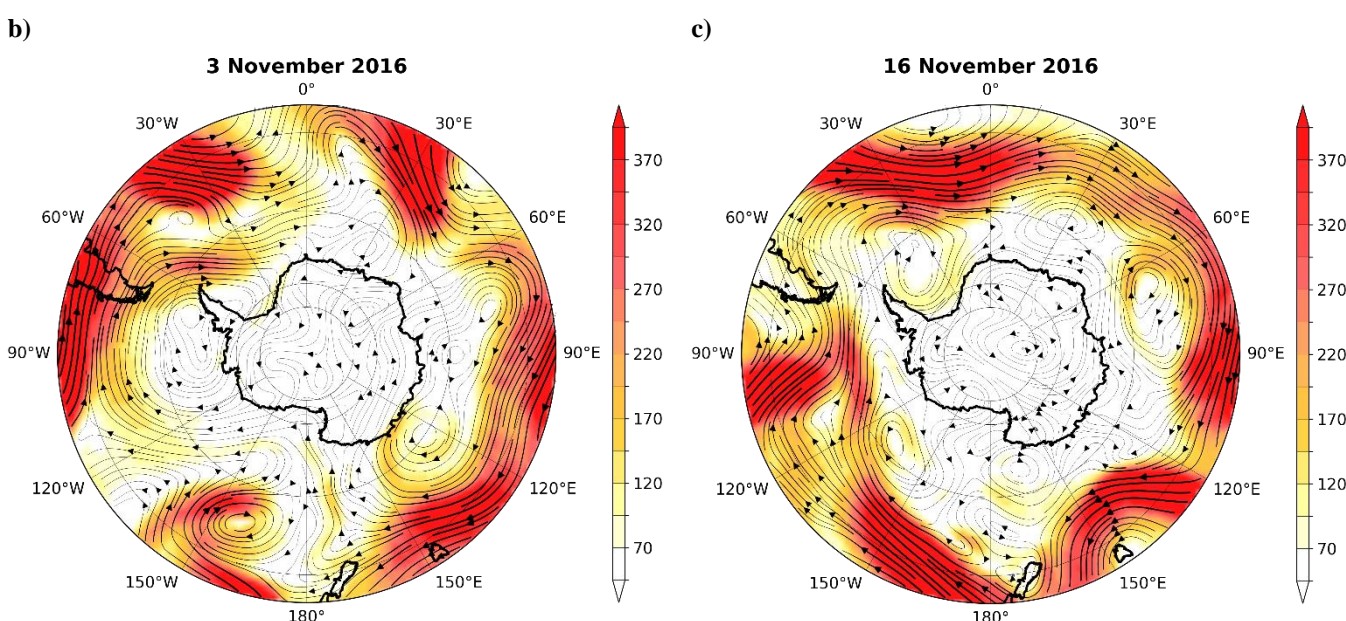

**Figure 10.** a) Daily IWT averaged over WS (black square in Figure 4h) over the period 1979 – 2016; b) The vertically integrated water vapor transport (WVT) on the $3^{rd}$ November 2016 and c) as in b) but for the $16^{th}$ November 2016. In b) and c) the color shaded areas indicated the magnitude of the moisture transport and the vectors the direction of the moisture transport.

*Table 1*. Rank of regional Sea Ice Extent from September to December 2016.

|        | September | October | November | December |
|--------|-----------|---------|----------|----------|
| ABS    | 5         | 4       | 8        | 4        |
| IO     | 2         | 2       | **1**    | 2        |
| RS     | 10        | 2       | **1**    | 2        |
| WP     | 15        | 11      | **1**    | 22       |
| WS     | 32        | 27      | 12       | **1**    |
| ANT    | 3         | 2       | **1**    | **1**    |

***Table 2***. Correlation coefficients between regional sea ice extent (SIE) and monthly Southern Annular Mode (SAM)

|  | **September** | **October** | **November** | **December** |
|---|---|---|---|---|
| **ABS** | 0.21 | 0.23 | 0.09 | 0.10 |
| **IO** | 0.07 | 0.17 | **0.43\*\*** | 0.31 |
| **RS** | -0.25 | -0.03 | 0.05 | **0.46\*\*** |
| **WPO** | 0.04 | 0.07 | 0.05 | 0.16 |
| **WS** | **-0.37\*** | -0.29 | -0.13 | 0.20 |

\* 95% significance level

\*\* 99% significance level