# Peer review of "Moisture transport and Antarctic sea ice: Austral spring 2016 event"

_Earth System Dynamics, 2017_

## Referee Comment (RC1) · Anonymous Referee #1 · 20 Feb 2018

**General Comments**

The dramatic decline of Antarctic sea ice in Austral Spring 2016 was in marked contrast to long-term positive trends, and the current authors join a small group of previous authors who have attempted to shed light on this. They provide some interesting additions, in particular the presentation of rank maps for understanding the long-term context, the use of lagged correlations with local climate variables (rather than indices) and the discussion of October preconditioning, and the consideration of moisture transport.

However, I judge that the paper in its current form does not provide sufficient insights or conclusions to merit publication. In particular, the main focus and novel feature of the paper is the analysis of water vapour transport but the authors do not justify (from prior work or physical arguments) why water vapour transport would be expected to impact sea ice, or demonstrate in their results that it gives novel information over temperature alone. In addition, the substantial long term context discussion on SIC and circulation overlaps with previous work and more acknowledgement of this is required. I am also not convinced that the presented analysis of the SAM adds value to the paper.

I do think that a substantially re-written paper building on the current moisture transport analysis could be a beneficial addition to the literature.

In terms of method, the stability maps method is clear; I would like to see some further justification of its robustness in the context of the short time series available in the Antarctic. More clarity is also needed on the water vapour variables used.

Regarding presentation, the structure of the paper is clear and the title and abstract are appropriate summaries of its content. However, the introduction does not appropriately introduce the paper. The discussion is also rather general, but I hope that if the authors make edits to the content of the paper, they will be able to draw some more specific conclusions in the discussion section.

The figures and captions also need some improvement.

All these points are detailed upon below.

**Specific Comments**

- Introduction: This is a general introduction to the importance of sea ice and to the differing observed and modelled behaviour observed at the two poles. It does not introduce the specific questions addressed in this paper. The authors need to introduce:

    o The work of previous authors (as already referenced in the discussion) on understanding the 2016 Austral Spring anomalies, and the gaps in these analyses which the authors address in this paper

    o Why they address moisture transport. Presumably the physical argument concerns downwelling longwave radiation? Some work has addressed this in the Arctic (e.g. Mortin et al, Yang and Magnusdottir) and it would be helpful to cite these.

    o The SAM and other circulation drivers and their relationship to sea ice

- P2, ~L12: 'In this respect': this is a slight leap from the argument for understanding physical processes behind the increase, to this paper addressing the dramatic opposite behaviour in Austral Spring 2016. Please reword.

- Data, P2L22: clarify that the linear interpolation is a standard part of the sea ice product (i.e. it is not an addition that you have made, therefore you do not need to justify it)

- Data, P3L2: How exactly is water vapour transport calculated from the ECMWF variables described, and which variables are used at which stage of the analysis?

- Data, P3L7: please expand on the reanalysis' performance as relevant to the current paper; e.g. Bracegirdle and Marshall showed this reanalysis gave good trend/variability of SLP and T at coastal locations. As far as I know there is not an evaluation of reanalysis IWV, due to lack of observations; perhaps clarify this as an irreducible uncertainty on your results?

- Data, P3L12: please give a brief description of the SAM and this SAM index.

- Methods, Stability Maps: The stability map method could be very beneficial in the Antarctic context. However, I am concerned about the shorter record (under 40 years compared to 100 years) and by necessity shorter moving windows (21 instead of 31) compared to your previous work. In particular, given 1979-2016 timeseries there must be 18 21-yr windows, none of them independent? Presumably this affects the interpretation of the stable correlations? Please comment. I was unsure about the defensibility of using 80% significance, but reassured by the fact you only pursue analysis where significance is over 95%; perhaps clarify this in the introduction to the method.

- Sections 3.1-3.3: the long term discussion overlaps significantly with previous work. More care is needed in citing these other papers (and perhaps shortening your description accordingly) e.g. BAMS state of the climate 2016 report sea ice section and relevant points from papers you already cite in the discussion. In particular at pg 5, line 2 (end of section 3.1) reference Turner et al 2017, who explicitly show the anomalous sea ice retreat in November 2016; and at page 5 line 20 cite BAMS state of the climate: Antarctica: Atmospheric circulation.

- P4 L24: I don't think WPO contributes notably to September SIE anomalies (Figure S2)

- P4 L27-28: Figure S2 shows WS does not become negative until the second half of November?

- P5 L23: I do not think you can argue the westerlies resulted in positive temperature anomalies; rather, all associated with same patterns of variability.

- Section 3.2:  Be careful in your use and discussion of the SAM, if retained.

    o It's not clear which behaviour you are saying 'projects onto the positive phase of the SAM'; the wavenumber 3 behaviour or the zonally symmetric annular structure. The Marshall index describes primarily zonally symmetric behaviour, whereas EOF based Antarctic Oscillation indices do capture the 3 centres of action.

    o I'm not sure it's useful; in some months (September and November) the circulation does have three centres of action in the expected places. However in October in particular it is not 'SAM-like' at all (either in the zonal-mean or zonally asymmetric sense). I would therefore suggest writing the discussion without reference to the SAM, and concluding the section with a short section, perhaps referencing a map of the SAM's typical behaviour (e.g. Spring SLP regressed onto the Marshall index, as a supplementary figure) and noting the months in which the circulation looked very SAM-like and the remarkable values of the index in this month. It would also be worth checking whether the SAM rankings are broadly robust to use of a different index.

- Section 3.3 and throughout: please ensure you are clear and consistent in your use of the terms 'water vapour', 'water vapour transport', 'total column water vapour',' integrated water vapour' etc and their associated acronyms. They were used inconsistently to the extent of my being unable to understand what was being used at all points

- Section 3.4: To me, the conclusion to be drawn from the maps and indices here (figs 5 and 6) is that there is a lagged local effect between temperature/water vapour and SIC in the regions analysed. This is perhaps not very surprising although I'm not aware of previous lagged analysis. Some questions to address to interpret the results:

    o To what extent is this a manifestation of persistence in SIE anomalies?

    o What are the results for the Weddell Sea in December (where anomalies are greater than in the Ross Sea)? 2016 saw rapid development of WS anomalies in November, implying that at least in this year, it was not just anomaly persistence.

    o Does IWV add more (statistical) information, or more physical understanding, over T alone? Figures 5 and 6 show the same regions in the stability maps for IWV and for T, and the timeseries look very highly correlated. A physical discussion and some supporting evidence is needed: is the same circulation bringing in heat and moisture? Or are water vapour anomalies radiatively driving temperature anomalies? Since this seems to me to be the main result of the paper, it is necessary to argue either that the water vapour has some independence from and therefore added predictive power over temperature, or that it adds physical understanding to the sea ice anomalies which cannot be inferred from temperature alone.

    o You discuss a dipole of stable regions. This could be related to the ASL, which would cause co-variability in the Ross and Amundsen-Bellingshausen sea, although the footprint in the Weddell sea is larger and further to the east than I'd expect in this case.

- P7L5: note that the highest SIE over this area is broadly true even when normalised anomalies are used (BAMS state of the climate figure 6.9); i.e. even accounting for natural variability these regions are exceptional in 2016.

- P7 L7; give the main results from the maps before discussing the indices.

- Section 3.5:

    o It is unclear which water vapour variables are used. Which figures show IWV and which show IWVT? Please check units, acronyms etc.

    o I found this hard to follow. I suggest rewriting this section such that each 2-day regional mini case study, is addressed with a little more care (maybe use 'first the ABS','second the IO','thirdly the RS'). Can you link these transport events to e.g. cyclones?

    o Take care over extrapolating to 'decline in first two weeks of December'. The anomalous decline in early December was in the WS (Fig S2) so I don't think you can robustly link it to the event shown.

- Section 3.6: This analysis does not add anything to the discussion about 2016, nor much to understanding of SAM-sea ice relationships in general. Table 2 is valuable but I think this discussion could be removed and replaced with a few sentences in the discussion e.g. 'Given the dramatic SAM anomalies in some months, we investigated the long term relationships between

regional SIE and the monthly SAM. However, significant relationships were found for only three month-region combinations (of 20) and thus consistent with previous studies […], we find the SAM's role in sea ice variability is complex.' The moving window method could be used to enhance this if the analysis is felt to be critical to the paper. Unless the relationship of the SAM to moisture transport is explicitly addressed, I do not think it adds to the novel scientific content of the paper.

- P9 L14; Is this lowest for November, or lowest overall? Is it true for other SAM indices?

- P9 L31; you say you've shown moisture and temperature anomalies could 'also' have led to different anomalies. 'Also' implies it's something different; are the anomalies you've shown not manifestations of the circulation anomalies discussed? 'poleward advection of warm': you don't show heat transport so this is a slight assumption, although I think Schlosser et al do- please cite.

- P10 L13 ; 'poleward advection of moist and warm air': increased moisture is not necessarily due to increased moisture advection?

- P10 L21; The study of Woods et al was about the Arctic. It's not clear it is relevant here.

**Technical Corrections**

- Abstract, Line 13: 'lowest daily sea ice concentration anomalies' -> 'largest magnitude negative daily sea ice concentration anomalies'

- Although it's almost certain readers know what SIE is, give the abbreviation at first mention (Pg 1 L25) or at first use in methods section (Pg 2 L23) [rather than at P2 L8 as done in this version].

- Page 1 Line 23: Artic -> Arctic

- P2, L7: Colins -> Collins

- P2, L11: clarify from offset what months 'Austral spring' is.

- P3, L20: on time-> in time

- P3, L21: Bracketed 'e.g.s' unnecessary and make text more confusing.

- P3, L31: Ionita et al (2017) reference should be Ionita (2017)?

- P4 : there is inconsistency between use of abbreviations ('WS') and full names ('weddell sea')

- P4 L4: 'problematic' a little emotive. Try 'challenging' or 'confusing'?

- P4 L5: '8'-> 'eight'

- P4 L6: 'positive SIE anomalies over the whole Antarctic region' -> 'positive pan-Antarctic SIE anomalies'. I don't like the phrase 'pan-Antarctic' much, but 'whole' implies 'everywhere'.

- P4 L20-21: rewrite e.g. 'In December the whole RS, and most of the ABS and IO were characterised by negative SIC anomalies'

- Pg 5 line 5: 'to the zonal' -> 'of zonal'.

- P5 L13: 'were'-> 'where'

- P5 L19: 'warming' -> 'warm anomalies'

- P6 L5: 'particular' -> 'exceptional'

- P6 L7: delete sentence 'The rank maps are computed…'

- P6 L10: 'to be able to clearly capture…2016 was' -> 'for clarity'

- P6 L13: delete 'Figures 3a,b and c indicate that'

- P6 L19, L23, L28: 'southern', 'eastern' and 'north-west' should be 'northern', 'western' and 'north-east'? Please check!

- P7L6, 16; delete 'respectively'

- P7 L9: 'considered'->'consider'

- P7 L24; 'December TT'-> 'December TT2'.

- P7 L27; delete 'a' before 'predictive'

- P8 L2 'there'->'three'

- P8 L3; 'employing' -> 'spatially averaging'?

- P8 L4 'region'->'regions'

- P8 L5: delete comma after 'IWV'

- P8 L10: reference Fig 7d

- P8 L13: 'BAS'->'ABS'.

- P8 L24: 'trimming'->'timing'.

- P8 L17: Reference Figure S2.

- P9 L15 'led to'->'associated with' (the SAM is in part expressing the behaviour of the westerlies. It is not a physical driver.).

- P9 L31 'lead'->'led'.

- P10 L16 'this'->'these'

- P10 L16 'moist'->'moisture'.

- P10 L21; 'In recent' ->'In a recent'.

**Corrections: Figures**

- Figure 1: Optional, but inclusion of topography and lines of latitude could make this a more useful reference figure for readers not as familiar with the Antarctic.

- Figure 2: The vectors are too small to be legible, and are not described in the caption. Repeating sector names in-plot is unnecessary here and the legibility is poor, so I suggest removing.

- Figure 3: As Figure 2 for sector labels. Remove '2016' in subfigure label titles. I also suggest you change the caption text to 'Ranking of 2016 monthly: sea ice concentration (SIC, first column, '1' means lowest SIC in period); integrated water vapour (IWV, second column, '1' means wettest

month in period); temperature (TT, third column, '1' means warmest month in period). The period is 1979-2016. Rankings below 5 appear white."

- o Technical question; how do repeated '0's rank (i.e. if several years have SIC=0?)

- o The colour scale is probably not colour-blind friendly. Please improve.

- Figure 5 and 6

    - o The correlations in e) and f) would be clearer if the SIE anomaly were negated.

    - o Add title to colour bar.

    - o Simplify caption text for panels a-d to (Fig 5 example): 'The stability maps between November Indian Ocean (IO) sea ice extent and a) October IWV, b) November IWV, c) October TT, d) November TT.

    - o Please check panel refs in caption to Fig 6; both time series panels referred to as c).

- Figure 7

    - o Check journal standards on colour scale (avoiding use of green-red for example).

    - o It would be helpful to show again in this figure the squares used to define regions.

    - o What are the vectors?

    - o Please clarify the dates shown in timeseries (are all days shown or only October and November) and check use and consistency of IWT/IWVT etc (see my general comments) and units.

- Supplementary Figure 1: In caption, mention 2016 just once, and state what the red number in each panel is (the ranking of that month's regional SIE).

- SF2: It would be helpful to include full regional names once more and to either change colours or find another way of making them clearer (IO, ANT and WS are very hard to distinguish) and of separating the full Antarctic time series from the individual regions. Y axis: Sie-> SIE.

- SF3: Interpolation of contours here is unhelpful as there is no propogation of anomalies in the x-direction. I suggest using a block rather than interpolated contour. Also please change colour scale and use less classes for legibility.

- SF4: See comments on (main text) Figure 3 caption and colour scale.

**References**

Mortin et al (2016): Melt onset over Arctic sea ice controlled by atmospheric moisture transport. DOI 10:1002/2016GL069330.

Yang and Magnusdottir (2017): Springtime extreme moisture transport into the Arctic and its impact on sea ice concentration. Doi 10.1002/2016JD026324.

---

## Referee Comment (RC2) · J.K. Kjellsson (Referee) · 18 Mar 2018

**Summary and overall comments**

The manuscript describes the dramatic retreat of AA sea ice in Oct-Dec 2016 in detail, and relates it to anomalies in the atmospheric circulation, in particular surface temperature, total column water vapour and total column water vapour flux. The methodology was previously unknown to me, and the results are interesting, warranting publication in ESD. However, the writing in the manuscript needs improvement. I therefore recommend publication after minor revisions.

**Specific comments**

Fig. 2: Please indicate in the caption what is shown. What are the arrows in left

column plots? In right column, are colours IMV (integrated water vapour) and arrows the IMVT (integrated water vapour transport)? Also, please insert reference arrow.

Fig 3: For comparison with Fig 2, please swap center and right columns so that temperature is in middle and IMV to the right.

Fig 4: This figure is referenced on lines 26, 29 on page 3, but I think you mean Fig 5 or 6? In that case, Fig 4 is only referenced to once in the paper, and is somewhat unnecessary. The relation between high temperatures and low sea ice is already made in Fig 3. I strongly suggest to remove it or add it to Supp. material.

Fig. 6: From Fig 6a,c, my guess is that there is an SLP anomaly somewhere over the Antarctic Peninsula driving the changes in advection. Hence, the 2nd box is correlated to WS sea ice, but is likely not causing the anomalies in WS sea ice? Also, the caption describes Fig 6c three times, but never Fig 6e or 6f.

Page 1, Line 11,12: Throughout the paper, the authors use the unusual abbreviations "IWV" for integrated water vapour and "TT" for 2m temperature. For the former, "VIWV" (vertically integrated water vapour) would be a better name so that the direction of integrations is explicit, or the authors could use "TCWV" (total column water vapour) or "IWVC" (integrated water vapour column) which are more common. Rather than "TT", perhaps "T2m" or "Ts" would be better?

Page 1, Line 32: The sentence "Antarctica experienced..." could be changed to "Between September 2016 and April 2017, Antarctica experienced a dramatic shift featuring strong negative SIE anomalies".

Line 2, Line 2: "could be caused either by a potential change..". What is meant by "potential change"? Perhaps re-phrase to "could potentially have been caused by a change in the long-term trend or by natural climate variability"?

Page 2, Line 20: Skip the comma between "utilize" and "the".
Page 3, Line 20: "does not change on time"? Do you mean "is time-independent"?

Section 2.2: The method is interesting, but the description may need some clarification. If there are 37 years of data and you use a 21-year moving window, then you only have 37-21=16 data points in time? Does this mean that the maps in Fig. 5,6 are made from 16 time levels? And 80% of 16 is about 13, so you require significant correlations for 13 of 16 time levels?

Page 3, Line 26, 29: Should be Figure 5,6 instead of 4,5?

Page 4, Line 22: Suggest "striking" instead of "very particular"

Page 4, Line 29, 30: Perhaps use "SIE anomaly below -2.2 mil. km2" instead of "more than 2.2" so that it is clear that the anomalies are negative. Same at Line 30.

Page 5, Line 2: "1980's"

Page 5, Line 12: At several places in the paper (e.g. Page 6, Line 32 and Page 7, Line 12), the authors use "positive temperature anomalies and enhanced water vapour". Why not use "positive anomalies in surface temperature and total column water vapour"?

Page 6, Lines 9, 15, 19, 20, 23: "wet" often refers to high precipitation. Use "most humid" instead? Also, do we know if these events are linked to high values of precipitation?

Page 6, Line 32: Suggest change from "extreme warm temperatures and enhanced moisture" to "large positive anomalies in surface temperature and total column water vapour".

Page 8, Lines 1-5: I found these sentences unclear. I would suggest "For October and November each year, we calculate daily IWVT (or TCWV or IWVC, see above comment) over three areas in the ABS, IO, RS respectively. The areas are those which rank as the warmest and most humid on record in 2016, and are marked in Figures 3e

ESDD
and 3h. The time series of IWVT for these areas are shown in Figures 7a, c, e. "

Page 8, Line 8: "reaching the" rather than "penetrating until".

Page 8, Line 13: "ABS" instead of "BAS"?

---

## Editor Comment (EC1) · A. Levermann (Editor) · 28 Mar 2018

Dear authors,

in your response, please bear the following in mind. The highly appreciated comment by the reviewer that he does not see how you made the connection between water vapor transport and sea ice extent, does NOT necessarily have to be met with additional analysis, but might just need further explanation why there is the connection that you claim. In general I think additional solid analysis towards a solution of a question that does not yet fully answer the question is still very helpful for the community and should be published.

Best wishes, Anders

---

## Author Comment (AC1) · 17 Apr 2018

We would like to thank the reviewer for the positive feedback on our manuscript and we are grateful for the comments on how it can be further improved. Here, we respond to each comment in turn – full details of the implementation will be provided in the revised manuscript.

General Comments

The dramatic decline of Antarctic sea ice in Austral Spring 2016 was in marked contrast to long-term positive trends, and the current authors join a small group of previous authors who have attempted to shed light on this. They provide some interesting additions, in particular the presentation of rank maps for understanding the long-term context, the use of lagged correlations with local climate variables (rather than indices) and the discussion of October preconditioning, and the consideration of moisture transport.

However, I judge that the paper in its current form does not provide sufficient insights or conclusions to merit publication. In particular, the main focus and novel feature of the paper is the analysis of water vapor transport but the authors do not justify (from prior work or physical arguments) why water vapor transport would be expected to impact sea ice, or demonstrate in their results that it gives novel information over temperature alone. In addition, the substantial long term context discussion on SIC and circulation overlaps with previous work and more acknowledgement of this is required. I am also not convinced that the presented analysis of the SAM adds value to the paper.

Concerning the first two paragraphs of the reviewer's comments we have some problems following the main arguments of the reviewer. In the first paragraph the reviewer states and we cite: "They provide some interesting additions, in particular the presentation of rank maps for understanding the long-term context, the use of lagged correlations with local climate variables (rather than indices) and the discussion of October preconditioning, and the consideration of moisture transport." In the following paragraph the reviewer states and we cite: "However, I judge that the paper in its current form does not provide sufficient insights or conclusions to merit publication. In particular, the main focus and novel feature of the paper is the analysis of water vapor transport but the authors do not justify (from prior work or physical arguments)…".

From our point of view these two paragraphs contradict each other. On one hand the reviewer does state that we bring new addition to this particular topic, on the other hand he/she states a lack of novelty.

We strongly demonstrated with the paper that we present new and novel insights regarding the topic under discussion. In the following we like to highlight them again in a short overview:

1. The rank maps. To our knowledge and due to intensive literature research, the performed study is the first one to make such an analysis and to put the 2016 event into a long term perspective both from a spatial and as well as from a temporal point of view.

2. The use of lagged correlations with climate fields, rather than pre-defined climate indices. To our knowledge and due to intensive litertature research, previous studies on this topic (e.g. Turner et al., 2017, Schlosser et al., 2017, Stoecker et al., 2017) have

**not employed lagged correlation analysis to study the relationship between the water vapor transport and Antarctic sea ice, at regional scale. The lagged analysis is definitely novel on this particular topic and our results indicate also that there is some predictive skill in the regional Antarctic sea ice from the previous month's background conditions.**

**3. The consideration of moisture transport. The inclusion of the moisture transport as a preconditioning for the development of Antarctic sea ice is again a novel idea in this context.**

**4. The stability maps. This is also a novel addition to the topic under discussion.**

**We strongly agree with the concerns of the reviewer regarding the water vapor transport and its impact on the sea ice, but we feel that this reason alone doesn't take away from the novelty of our analysis. In the revised version of the manuscript we will extend the discussion regarding the influence of moisture transport on the sea ice and performed additional work on actual literature. Thus we will add new references. We also add new figures to strengthen our findings and to underline the novelty and importance of our paper.**

**Following the reviewer's comments we will shorten the section regarding the long term context discussion on SIC and large-scale atmospheric circulation as well as the SAM part. Following the suggestions we will focus more on the water vapor transport. The according references will be added and acknowledged in a appropriate manner.**

I do think that a substantially re-written paper building on the current moisture transport analysis could be a beneficial addition to the literature.

**As stated above, in the revised version of the manuscript we will make a more detailed discussion about the moisture transport analysis and its impact on the Antarctic sea ice. Therefore we will fulfill this suggestion of the reviewer in the revised version.**

In terms of method, the stability maps method is clear; I would like to see some further justification of its robustness in the context of the short time series available in the Antarctic. More clarity is also needed on the water vapor variables used.

**This good suggestion of the reviewer will be implemented in the revised version of the manuscript. A justification regarding the use of stability maps and their robustness with short time series will be added.**

Regarding presentation, the structure of the paper is clear and the title and abstract are appropriate summaries of its content. However, the introduction does not appropriately introduce the paper. The discussion is also rather general, but I hope that if the authors make edits to the content of the paper, they will be able to draw some more specific conclusions in the discussion section.

**Following the reviewer's suggestion we will rewrite the 'Introduction part' and the discussion and conclusions part.**

The figures and captions also need some improvement.
All these points are detailed upon below.

**Specific Comments**

**All the specific comments of the reviewer will be taken into account in the revised version of the manuscript. In the following we will answer just to some of these comments, but each comment/suggestion of the reviewer will be considered and integrated in the revised version of the manuscript.**

Introduction: This is a general introduction to the importance of sea ice and to the differing observed and modelled behavior observed at the two poles. It does not introduce the specific questions addressed in this paper. The authors need to introduce:

o The work of previous authors (as already referenced in the discussion) on understanding the 2016 Austral Spring anomalies, and the gaps in these analyses which the authors address in this paper

**This will be included in the revised version of the paper. Missing references will be added and the questions addressed in the paper will be introduced more precisely.**

o Why they address moisture transport. Presumably the physical argument concerns downwelling longwave radiation? Some work has addressed this in the Arctic (e.g. Mortin et al, Yang and Magnusdottir) and it would be helpful to cite these.

**This will be clarified and the missing references will be added. More work will be performed to state this point out more precisely.**

o The SAM and other circulation drivers and their relationship to sea ice

P2, ~L12: 'In this respect': this is a slight leap from the argument for understanding physical processes behind the increase, to this paper addressing the dramatic opposite behavior in Austral Spring 2016. Please reword.

**This sentences will be reworded and clarified.**

Data, P2L22: clarify that the linear interpolation is a standard part of the sea ice product (i.e. it is not an addition that you have made, therefore you do not need to justify it)

**This will be rephrased accordingly.**

Data, P3L2: How exactly is water vapor transport calculated from the ECMWF variables described, and which variables are used at which stage of the analysis?

**This information will be added in the revised version of the paper. Furthermore, we will state out more precisely which variable is used at which stage of the analysis.**

Data, P3L7: please expand on the reanalysis' performance as relevant to the current paper; e.g. Bracegirdle and Marshall showed this reanalysis gave good trend/variability of SLP and T at coastal locations. As far as I know there is not an evaluation of reanalysis IWV, due to lack of observations; perhaps clarify this as an irreducible uncertainty on your results?

**We agree with the reviewer's concern regarding the uncertainty of the ECMWF data over Antarctica. We will try to add more information regarding this issue in the revised version of the manuscript.**

Data, P3L12: please give a brief description of the SAM and this SAM index.

**This will be done and it will be also added some recent references.**

Methods, Stability Maps: The stability map method could be very beneficial in the Antarctic context. However, I am concerned about the shorter record (under 40 years compared to 100 years) and by necessity shorter moving windows (21 instead of 31) compared to your previous work. In particular, given 1979-2016 time series there must be 21-yr windows, none of them independent? Presumably this affects the interpretation of the stable correlations? Please comment. I was unsure about the defensibility of using 80% significance, but reassured by the fact you only pursue analysis where significance is over 95%; perhaps clarify this in the introduction to the method.

**An analysis and justification regarding the use of stability maps and their robustness with short time series will be added in the revised version of the manuscript. By this the questions of the reviewer should be answered. This will surely help to make the method applied for Antarctic time series more understandable.**

Sections 3.1-3.3: the long term discussion overlaps significantly with previous work. More care is needed in citing these other papers (and perhaps shortening your description accordingly) e.g. BAMS state of the climate 2016 report sea ice section and relevant points from papers you already cite in the discussion. In particular at pg 5, line 2 (end of section 3.1) reference Turner et al 2017, who explicitly show the anomalous sea ice retreat in November 2016; and at page 5 line 20 cite BAMS state of the climate: Antarctica: Atmospheric circulation.

**This is a very helpful suggestion to strengthen the references and work already done and not yet clearly be presented in the paper. This will be realized in the revised version of the paper.**

P4 L24: I don't think WPO contributes notably to September SIE anomalies (Figure S2)

**The text will be changed accordingly.**

P4 L27-28: Figure S2 shows WS does not become negative until the second half of November?

**The text will be changed accordingly.**

P5 L23: I do not think you can argue the westerlies resulted in positive temperature anomalies; rather, all associated with same patterns of variability.

**These two sections will be shortened and a more detailed analysis regarding the water vapor transport will be added in the revised version of the manuscript. This will help to clarify this comment.**

Section 3.2: Be careful in your use and discussion of the SAM, if retained.

o It's not clear which behaviour you are saying 'projects onto the positive phase of the SAM'; the wavenumber 3 behaviour or the zonally symmetric annular structure. The Marshall index describes primarily zonally symmetric behaviour, whereas EOF based Antarctic Oscillation indices do capture the 3 centres of action.

o I'm not sure it's useful; in some months (September and November) the circulation does have three centres of action in the expected places. However in October in particular it is not 'SAM-like' at all (either in the zonal-mean or zonally asymmetric sense). I would therefore suggest writing the discussion without reference to the SAM, and concluding the section with a short section, perhaps referencing a map of the SAM's typical behaviour (e.g. Spring SLP regressed onto the Marshall index, as a supplementary figure) and noting the months in which the circulation looked very SAM-like and the remarkable values of the index in this month. It would also be worth checking whether the SAM rankings are broadly robust to use of a different index.

**We agree with the reviewer's comment regarding the two paragraphs above. Based on the aforementioned comments, we will rephrase the content of this section of the paper and add also the ranking for some based on a different SAM index.**

Section 3.3 and throughout: please ensure you are clear and consistent in your use of the terms 'water vapour', 'water vapour transport', 'total column water vapour',' integrated water vapour' etc and their associated acronyms. They were used inconsistently to the extent of my being unable to understand what was being used at all points

**We fully agree with this comment. We will change the terminology to be more clear and consistent. By this the paper will be become much better readable.**

Section 3.4: To me, the conclusion to be drawn from the maps and indices here (figs 5 and 6) is that there is a lagged local effect between temperature/water vapour and SIC in the regions analyzed. This is perhaps not very surprising although I'm not aware of previous lagged analysis. Some questions to address to interpret the results:

o To what extent is this a manifestation of persistence in SIE anomalies?

**Persistence plays definitely a role, but without a strong atmospheric/oceanic forcing an event like 2016 is rather impossible to occur. In the revised version of the manuscript we will try to discuss also the influence of persistence on the development of the SIE anomalies throughout this particular event.**

o What are the results for the Weddell Sea in December (where anomalies are greater than in the Ross Sea)? 2016 saw rapid development of WS anomalies in November, implying that at least in this year, it was not just anomaly persistence.

**We will check the results for Weddell Sea in December and we will include them if important findings will be found.**

o Does IWV add more (statistical) information, or more physical understanding, over T alone? Figures 5 and 6 show the same regions in the stability maps for IWV and for T, and the timeseries look very highly correlated. A physical discussion and some supporting evidence is needed: is the same circulation bringing in heat and moisture? Or are water vapour anomalies

radiatively driving temperature anomalies? Since this seems to me to be the main result of the paper, it is necessary to argue either that the water vapour has some independence from and therefore added predictive power over temperature, or that it adds physical understanding to the sea ice anomalies which cannot be inferred from temperature alone.

**We are thankful for this comments as they help to make the paper's arguments more consistent and better understandable. A new figure and text will be added, on which we will try to quantify the contribution of the water vapor transport and temperature to the regional sea ice.**

o You discuss a dipole of stable regions. This could be related to the ASL, which would cause co-variability in the Ross and Amundsen-Bellingshausen sea, although the footprint in the Weddell sea is larger and further to the east than I'd expect in this case.

**ASL could play a role, at least partially, but due to the location of the centers of the dipole we feel that a general discussion is more appropriate compared to relating this just the ASL variability. As already commented by the reviewer, the footprint in the Weddell Sea is rather different compared to the one observed for ASL.**

P7L5: note that the highest SIE over this area is broadly true even when normalized anomalies are used (BAMS state of the climate figure 6.9); i.e. even accounting for natural variability these regions are exceptional in 2016.

**As reference to the BAMS state of the climate will be added.**

P7 L7; give the main results from the maps before discussing the indices.

**We will take this suggestion into account when we revise the paper.**

Section 3.5:
o It is unclear which water vapour variables are used. Which figures show IWV and which show IWVT? Please check units, acronyms etc.

**The text will be modified following these suggestions to make it more clear and easy to follow. All the units and acronyms will be carefully checked.**

o I found this hard to follow. I suggest rewriting this section such that each 2-day regional mini case study, is addressed with a little more care (maybe use 'first the ABS',' second the IO',' thirdly the RS'). Can you link these transport events to e.g. cyclones?

**In the revised version of the manuscript we will rewrite the section to make it more clear and easy to follow.**

o Take care over extrapolating to 'decline in first two weeks of December'. The anomalous decline in early December was in the WS (Fig S2) so I don't think you can robustly link it to the event shown.

**The text will be modified following the reviewer's suggestions.**

Section 3.6: This analysis does not add anything to the discussion about 2016, nor much to understanding of SAM-sea ice relationships in general. Table 2 is valuable but I think this

discussion could be removed and replaced with a few sentences in the discussion e.g. 'Given the dramatic SAM anomalies in some months, we investigated the long term relationships between regional SIE and the monthly SAM. However, significant relationships were found for only three month-region combinations (of 20) and thus consistent with previous studies […], we find the SAM's role in sea ice variability is complex.' The moving window method could be used to enhance this if the analysis is felt to be critical to the paper. Unless the relationship of the SAM to moisture transport is explicitly addressed, I do not think it adds to the novel scientific content of the paper.

P9 L14; Is this lowest for November, or lowest overall? Is it true for other SAM indices?

**As stated above, the discussion regarding SAM will be shortened and changed following the suggestions made by the reviewer.**

P9 L31; you say you've shown moisture and temperature anomalies could 'also' have led to different anomalies. 'Also' implies it's something different; are the anomalies you've shown not manifestations of the circulation anomalies discussed? 'poleward advection of warm': you don't show heat transport so this is a slight assumption, although I think Schlosser et al do-please cite.

**At the submission date of our paper to ESD (17.11.2017) in the Schlosser et al paper (under discussion at that time) there were no figures regarding the heat transport. Nevertheless, the final version of Schlosser et al paper has been recently published, with the addition of the heat transport figure, and now the paper will be cited and referred accordingly.**

P10 L13 ; 'poleward advection of moist and warm air': increased moisture is not necessarily due to increased moisture advection?

**We agree with the reviewer's comment and we will modify the text accordingly.**

P10 L21; The study of Woods et al was about the Arctic. It's not clear it is relevant here.

**The aforementioned reference will be deleted.**

**Technical Corrections**

Abstract, Line 13: 'lowest daily sea ice concentration anomalies' -> 'largest magnitude negative daily sea ice concentration anomalies'
Although it's almost certain readers know what SIE is, give the abbreviation at first mention (Pg 1 L25) or at first use in methods section (Pg 2 L23) [rather than at P2 L8 as done in this version].
Page 1 Line 23: Artic -> Arctic
P2, L7: Colins -> Collins
P2, L11: clarify from offset what months 'Austral spring' is.
P3, L20: on time-> in time
P3, L21: Bracketed 'e.g.s' unnecessary and make text more confusing.
P3, L31: Ionita et al (2017) reference should be Ionita (2017)?
P4 : there is inconsistency between use of abbreviations ('WS') and full names ('weddell sea')
P4 L4: 'problematic' a little emotive. Try 'challenging' or 'confusing'?

P4 L5: '8'-> 'eight'

P4 L6: 'positive SIE anomalies over the whole Antarctic region' -> 'positive pan-Antarctic SIE anomalies'. I don't like the phrase 'pan-Antarctic' much, but 'whole' implies 'everywhere'.

P4 L20-21: rewrite e.g. 'In December the whole RS, and most of the ABS and IO were characterised by negative SIC anomalies'

Pg 5 line 5: 'to the zonal' -> 'of zonal'.

P5 L13: 'were'-> 'where'

P5 L19: 'warming' -> 'warm anomalies'

P6 L5: 'particular' -> 'exceptional'

P6 L7: delete sentence 'The rank maps are computed…'

P6 L10: 'to be able to clearly capture…2016 was' -> 'for clarity'

P6 L13: delete 'Figures 3a,b and c indicate that'

P6 L19, L23, L28: 'southern', 'eastern' and 'north-west' should be 'northern', 'western' and 'north-east'? Please check!

P7L6, 16; delete 'respectively'

P7 L9: 'considered'->'consider'

P7 L24; 'December TT'-> 'December TT2'.

P7 L27; delete 'a' before 'predictive'

P8 L2 'there'->'three'

P8 L3; 'employing' -> 'spatially averaging'?

P8 L4 'region'->'regions'

P8 L5: delete comma after 'IWV'

P8 L10: reference Fig 7d

P8 L13: 'BAS'->'ABS'.

P8 L24: 'trimming'->'timing'.

P8 L17: Reference Figure S2.

P9 L15 'led to'->'associated with' (the SAM is in part expressing the behaviour of the westerlies. It is not a physical driver.).

P9 L31 'lead'->'led'.

P10 L16 'this'->'these'

P10 L16 'moist'->'moisture'.

P10 L21; 'In recent' ->'In a recent'.

**All the technical corrections will be addressed and inserted in the revised version of the manuscript. We are thankful for the very careful reading of the reviewer at this point.**

Corrections: Figures

Figure 1: Optional, but inclusion of topography and lines of latitude could make this a more useful reference figure for readers not as familiar with the Antarctic.

**We will try to include a better figure to comply with the reviewer's suggestion.**

Figure 2: The vectors are too small to be legible, and are not described in the caption. Repeating sector names in-plot is unnecessary here and the legibility is poor, so I suggest removing.

**The figure will be changed according to the suggestions.**

Figure 3: As Figure 2 for sector labels. Remove '2016' in subfigure label titles. I also suggest you change the caption text to 'Ranking of 2016 monthly: sea ice concentration (SIC, first column, '1' means lowest SIC in period); integrated water vapour (IWV, second column, '1' means wettest month in period); temperature (TT, third column, '1' means warmest month in period). The period is 1979-2016. Rankings below 5 appear white."

**The aforementioned suggestions will be taken into account and the figure caption for Figure 2 will be modified accordingly.**

o Technical question; how do repeated '0's rank (i.e. if several years have SIC=0?)

**The years with SIC = 0 are treated as NaN's and they are not taken into account.**

o The colour scale is probably not colour-blind friendly. Please improve. Figure 5 and 6

**We will try to find a better color scheme to avoid the color-blind issue.**

o The correlations in e) and f) would be clearer if the SIE anomaly were negated.

**We will reverse to SIE anomaly for a clearer view.**

o Add title to colour bar.

**Will be done.**

o Simplify caption text for panels a-d to (Fig 5 example): 'The stability maps between November Indian Ocean (IO) sea ice extent and a) October IWV, b) November IWV, c) October TT, d) November TT.

We will follow the suggestions and simplify the caption text in order to make them more understandable.

**All following suggestions will be implemented in the revised version of the paper.**

o Please check panel refs in caption to Fig 6; both time series panels referred to as c). Figure 7
o Check journal standards on colour scale (avoiding use of green-red for example).
o It would be helpful to show again in this figure the squares used to define regions.
o What are the vectors?
o Please clarify the dates shown in timeseries (are all days shown or only October and November) and check use and consistency of IWT/IWVT etc (see my general comments) and units.

Supplementary Figure 1: In caption, mention 2016 just once, and state what the red number in each panel is (the ranking of that month's regional SIE).
SF2: It would be helpful to include full regional names once more and to either change colours or find another way of making them clearer (IO, ANT and WS are very hard to

distinguish) and of separating the full Antarctic time series from the individual regions. Y axis: Sie-> SIE.

SF3: Interpolation of contours here is unhelpful as there is no propagation of anomalies in the x-direction. I suggest using a block rather than interpolated contour. Also please change colour scale and use less classes for legibility.

SF4: See comments on (main text) Figure 3 caption and colour scale.

References

Mortin et al (2016): Melt onset over Arctic sea ice controlled by atmospheric moisture transport. DOI 10:1002/2016GL069330.

Yang and Magnusdottir (2017): Springtime extreme moisture transport into the Arctic and its impact on sea ice concentration. Doi 10.1002/2016JD026324.

**All the figures will be modified following the reviewer's suggestion.**

---

## Author Comment (AC2) · 17 Apr 2018

**We would like to thank the reviewer for the positive feedback on our manuscript and we are grateful for the comments on how it can be further improved. Here, we respond to each comment in turn – full details of the implementation will be provided in the revised manuscript.**

**Summary and overall comments**

The manuscript describes the dramatic retreat of AA sea ice in Oct-Dec 2016 in detail, and relates it to anomalies in the atmospheric circulation, in particular surface temperature, total column water vapour and total column water vapour flux. The methodology was previously unknown to me, and the results are interesting, warranting publication in ESD. However, the writing in the manuscript needs improvement. I therefore recommend publication after minor revisions.

**Specific comments**

Fig. 2: Please indicate in the caption what is shown. What are the arrows in left column plots? In right column, are colours IMV (integrated water vapour) and arrows the IMVT (integrated water vapour transport)? Also, please insert reference arrow.

Fig 3: For comparison with Fig 2, please swap center and right columns so that temperature is in middle and IMV to the right.

Fig 4: This figure is referenced on lines 26, 29 on page 3, but I think you mean Fig 5 or 6? In that case, Fig 4 is only referenced to once in the paper, and is somewhat unnecessary. The relation between high temperatures and low sea ice is already made in Fig 3. I strongly suggest to remove it or add it to Supp. material.

Fig. 6: From Fig 6a,c, my guess is that there is an SLP anomaly somewhere over the Antarctic Peninsula driving the changes in advection. Hence, the 2nd box is correlated to WS sea ice, but is likely not causing the anomalies in WS sea ice? Also, the caption describes Fig 6c three times, but never Fig 6e or 6f.

**All the figures will be modified/improved following the reviewer's suggestion.**

Page 1, Line 11,12: Throughout the paper, the authors use the unusual abbreviations "IWV" for integrated water vapour and "TT" for 2m temperature. For the former, "VIWV" (vertically integrated water vapour) would be a better name so that the direction of integrations is explicit, or the authors could use "TCWV" (total column water vapour) or "IWVC" (integrated water vapour column) which are more common. Rather than "TT", perhaps "T2m" or "Ts" would be better?

**We strongly agree with this issue, and we will try to find better wording/abbreviations to make the paper easy to read. The text will be checked for inconsistencies and modified accordingly.**

Page 1, Line 32: The sentence "Antarctica experienced..." could be changed to "Between September 2016 and April 2017, Antarctica experienced a dramatic shift featuring strong negative SIE anomalies".

**The text will be modified accordingly.**

Line 2, Line 2: "could be caused either by a potential change..". What is meant by "potential change"? Perhaps re-phrase to "could potentially have been caused by a change in the long-term trend or by natural climate variability"?

**The text will be rephrased following the reviewer's suggestion.**

Page 2, Line 20: Skip the comma between "utilize" and "the".

Page 3, Line 20: "does not change on time"? Do you mean "is time-independent"?

**The text will be rephrased following the reviewer's suggestion.**

Section 2.2: The method is interesting, but the description may need some clarification. If there are 37 years of data and you use a 21-year moving window, then you only have 37-21=16 data points in time? Does this mean that the maps in Fig. 5,6 are made from 16 time levels? And 80% of 16 is about 13, so you require significant correlations for 13 of 16 time levels?

**A justification regarding the use of stability maps and their robustness with short time series will be added in the revised version of the manuscript.**

Page 3, Line 26, 29: Should be Figure 5,6 instead of 4,5?

Page 4, Line 22: Suggest "striking" instead of "very particular"

Page 4, Line 29, 30: Perhaps use "SIE anomaly below −2.2 mil. km2" instead of "more than 2.2" so that it is clear that the anomalies are negative. Same at Line 30.

Page 5, Line 2: "1980's"

Page 5, Line 12: At several places in the paper (e.g. Page 6, Line 32 and Page 7, Line 12), the authors use "positive temperature anomalies and enhanced water vapour". Why not use "positive anomalies in surface temperature and total column water vapour"?

Page 6, Lines 9, 15, 19, 20, 23: "wet" often refers to high precipitation. Use "most humid" instead? Also, do we know if these events are linked to high values of precipitation?

Page 6, Line 32: Suggest change from "extreme warm temperatures and enhanced moisture" to "large positive anomalies in surface temperature and total column water vapour".

Page 8, Lines 1-5: I found these sentences unclear. I would suggest "For October and November each year, we calculate daily IWVT (or TCWV or IWVC, see above comment) over three areas in the ABS, IO, RS respectively. The areas are those which rank as the warmest and most humid on record in 2016, and are marked in Figures 3e and 3h. The time series of IWVT for these areas are shown in Figures 7a, c, e. "

Page 8, Line 8: "reaching the" rather than "penetrating until".

Page 8, Line 13: "ABS" instead of "BAS"?

**All the technical issues and the suggestions made by the reviewer (see comments above) will be fixed in the revised version of the manuscript.**

---

## Author Response (AR1)

Response Reviewer 1

We would like to thank the reviewer for the feedback on our manuscript and we are grateful for the comments on how it can be further improved. We provide below a point by point response to the reviewer comments/suggestions.

General Comments
The dramatic decline of Antarctic sea ice in Austral Spring 2016 was in marked contrast to long-term positive trends, and the current authors join a small group of previous authors who have attempted to shed light on this. They provide some interesting additions, in particular the presentation of rank maps for understanding the long-term context, the use of lagged correlations with local climate variables (rather than indices) and the discussion of October preconditioning, and the consideration of moisture transport.

However, I judge that the paper in its current form does not provide sufficient insights or conclusions to merit publication. In particular, the main focus and novel feature of the paper is the analysis of water vapor transport but the authors do not justify (from prior work or physical arguments) why water vapor transport would be expected to impact sea ice, or demonstrate in their results that it gives novel information over temperature alone. In addition, the substantial long term context discussion on SIC and circulation overlaps with previous work and more acknowledgement of this is required. I am also not convinced that the presented analysis of the SAM adds value to the paper.

Concerning the first two paragraphs of the reviewer's comments we see a discrepancy in the reviewer comments and so we have some problems following the main arguments of the reviewer. In the first paragraph the reviewer states and we cite: "They provide some interesting additions, in particular the presentation of rank maps for understanding the long-term context, the use of lagged correlations with local climate variables (rather than indices) and the discussion of October preconditioning, and the consideration of moisture transport." In the following paragraph the reviewer states and we cite: "However, I judge that the paper in its current form does not provide sufficient insights or conclusions to merit publication. In particular, the main focus and novel feature of the paper is the analysis of water vapor transport but the authors do not justify (from prior work or physical arguments)…".

From our point of view these two paragraphs contradict each other. On the one hand the reviewer states that we bring new addition to this particular topic, on the other hand the reviewer states a lack of novelty.

In our opinion we strongly demonstrated with this paper that we present new and novel insights regarding the topic under discussion, namely an exceptional anomalous long-term sea ice retreat in austral spring 2016. In the following we like to highlight them again in a short overview:

1. The rank maps. To our knowledge and due to intensive literature research, the performed study is the first one making such an analysis on sea ice extent and putting the 2016 event into a long term perspective both from a spatial and as well as from a temporal point of view.

2. The use of lagged correlations with climate fields, rather than pre-defined climate indices. To our knowledge and supported by an intensive literature research, previous studies on this topic (e.g. Turner et al., 2017, Schlosser et al., 2017, Stoecker et al., 2017) have not employed lagged correlation analysis to study the relationship between the water vapor transport and Antarctic sea ice, at regional scale. The lagged analysis is definitely novel on this particular topic and our results indicate also that there is some predictive skill in the regional Antarctic sea ice from the previous month's background conditions.

3. The consideration of moisture transport. The inclusion of the moisture transport as a preconditioning for the development of Antarctic sea ice is again a novel idea in this context.

4. The stability maps. This is also a novel addition to the topic under discussion.

We agree with the concerns of the reviewer that the discussion of the water vapor transport and its impact on the sea ice needs further elaboration and argumentation, but we feel that this reason alone doesn't take away the novelty of our analysis. Therefore, in the revised version of the manuscript we have extended the discussion regarding the influence of moisture transport on the sea ice and performed additional work on actual literature, adding new and additional references. We also improved/added new figures (Figure 3, 9 and 10) to strengthen our findings and to underline the novelty and importance of our paper.

Following the reviewer's comments we have removed the SAM part (former Section 3.6). Furthermore, we have focused the revised version of our manuscript more on the water vapor transport as suggested. The according references have been added and acknowledged in an appropriate manner.

I do think that a substantially re-written paper building on the current moisture transport analysis could be a beneficial addition to the literature.

As stated above, in the revised version of the manuscript, we have added a more detailed discussion about the moisture transport analysis and its impact on the Antarctic sea ice (see Introduction, Section 3.5, and the Discussion part). This discussion was elaborated and extended according to the suggestions.

In terms of method, the stability maps method is clear; I would like to see some further justification of its robustness in the context of the short time series available in the Antarctic. More clarity is also needed on the water vapor variables used.

This good suggestion of the reviewer has been implemented in the revised version of the manuscript. A discussion regarding the use of stability maps and their robustness with short time series has been added for clarification (Section 2.2 – Methods).

Regarding presentation, the structure of the paper is clear and the title and abstract are appropriate summaries of its content. However, the introduction does not appropriately introduce the paper. The discussion is also rather general, but I hope that if the authors make edits to the content of the paper, they will be able to draw some more specific conclusions in the discussion section.

The "Introduction" and "Discussion and conclusions" parts have been re-written following the reviewer's suggestions. By this we believe that the both sections now are straight forward regarding the framing of the work as well as in putting the results in a proper context.

The figures and captions also need some improvement.
All these points are detailed upon below.

**Specific Comments**

All the specific comments of the reviewer haven been taken into account in the revised version of the manuscript. In the following we will answer how each of the specific comments have been considered I the revised version of the manuscript.

Introduction: This is a general introduction to the importance of sea ice and to the differing observed and modelled behavior observed at the two poles. It does not introduce the specific questions addressed in this paper. The authors need to introduce:

o The work of previous authors (as already referenced in the discussion) on understanding the 2016 Austral Spring anomalies, and the gaps in these analyses which the authors address in this paper

The work of previous authors has been added in the revised version of the manuscript. (page 2, lines 15 – 23). New references have been also added in the revised version of the manuscript.

o Why they address moisture transport. Presumably the physical argument concerns downwelling longwave radiation? Some work has addressed this in the Arctic (e.g. Mortin et al, Yang and Magnusdottir) and it would be helpful to cite these.

We have discussed this extensively issues on page 2 (lines 24 – 35) and page 3 (lines 1 – 3) of the revised version of the manuscript.

"Due to the fact that the Arctic and Antarctic regions are moisture flux convergence areas, atmospheric moisture transport is a primary input of water into these regions. Moreover, through the cloud-radiative forcing the moisture transport directly or indirectly affects the snow, sea ice and ice sheet over the polar regions. For the Arctic region, Kapsch et al. (2013) demonstrated that a significantly enhanced transport of humid air during spring produces increased cloudiness and humidity, thus accelerating the sea ice retreat in summer. Therefore, atmospheric moisture transport is a crucial component for the water balance, especially over the polar regions. While for the Arctic regions there is a large number of studies dealing with the influence of the moisture transport on the sea ice variability (Mortin et al., 2016; Yang and Magnusdottir, 2017; Park et al., 20125), over the Antarctic region little attention has been paid to transport variations of moisture from the Extratropics (Nieto et al., 2017). As such, the objectives of our paper are as follows: (a) to characterize the temporal and spatial extent of the spring 2016 exceptional sea ice melting event using both daily and monthly sea ice data; (b) to analyze the key drivers of the event, with a special emphasis on the role played by enhanced moisture transport and warm intrusions into the Antarctic region; (c) to place the austral spring 2016 into a long-term perspective. The

paper is structured as follows: in Section 2, we introduce the data used in this study; the main results of our analysis are shown in Section 3, while the concluding remarks are presented in Section 4."

o The SAM and other circulation drivers and their relationship to sea ice

Most of the SAM analysis has been removed from the revised version of the manuscript.

P2, ~L12: 'In this respect': this is a slight leap from the argument for understanding physical processes behind the increase, to this paper addressing the dramatic opposite behavior in Austral Spring 2016. Please reword.

This sentence has been reworded and clarified.

Data, P2L22: clarify that the linear interpolation is a standard part of the sea ice product (i.e. it is not an addition that you have made, therefore you do not need to justify it)

This has been rephrased accordingly.

Data, P3L2: How exactly is water vapor transport calculated from the ECMWF variables described, and which variables are used at which stage of the analysis?

This information has been added in the revised version of the paper. Furthermore, we have stated out more precisely which variable is used at which stage of the analysis.

Data, P3L7: please expand on the reanalysis' performance as relevant to the current paper; e.g. Bracegirdle and Marshall showed this reanalysis gave good trend/variability of SLP and T at coastal locations. As far as I know there is not an evaluation of reanalysis IWV, due to lack of observations; perhaps clarify this as an irreducible uncertainty on your results?

We agree with the reviewer's concern regarding the uncertainty of the ECMWF data over Antarctica. We have tried to add more information regarding this issue in the revised version of the manuscript (see Section 2.1 Data) and added some new references to frame this issue.

Data, P3L12: please give a brief description of the SAM and this SAM index.

This is done and new references are included

Methods, Stability Maps: The stability map method could be very beneficial in the Antarctic context. However, I am concerned about the shorter record (under 40 years compared to 100 years) and by necessity shorter moving windows (21 instead of 31) compared to your previous work. In particular, given 1979-2016 time series there must be 21-yr windows, none of them independent? Presumably this affects the interpretation of the stable correlations? Please comment. I was unsure about the defensibility of using 80% significance, but reassured by the fact you only pursue analysis where significance is over 95%; perhaps clarify this in the introduction to the method.

An analysis and justification regarding the use of stability maps and their robustness with short time series has been added in the revised version of the manuscript. By this the questions of the reviewer should be answered. This will surely help to make the method applied for Antarctic time series more understandable. (see section 2.2 Methods).

Sections 3.1-3.3: the long term discussion overlaps significantly with previous work. More care is needed in citing these other papers (and perhaps shortening your description accordingly) e.g. BAMS state of the climate 2016 report sea ice section and relevant points from papers you already cite in the discussion. In particular at pg 5, line 2 (end of section 3.1) reference Turner et al 2017, who explicitly show the anomalous sea ice retreat in November 2016; and at page 5 line 20 cite BAMS state of the climate: Antarctica: Atmospheric circulation.

This is a very helpful suggestion to strengthen the references and work already done and not yet clearly presented in the paper. Nevertheless, the overlapping discussion with the other papers is fairly low  and we have acknowledged the contribution of the other studies in the revised version of the manuscript. Without having a short discussion about some of the atmospheric drivers (e.g. SLP and air temperature) it would have became almost impossible for us to have a clear story regarding the 2016 event.

P4 L24: I don't think WPO contributes notably to September SIE anomalies (Figure S2)

The text has been changed accordingly in order to have the proper statement in the paper.

P4 L27-28: Figure S2 shows WS does not become negative until the second half of November?

The text has been changed accordingly.

 P5 L23: I do not think you can argue the westerlies resulted in positive temperature anomalies; rather, all associated with same patterns of variability.

The text has been modified.

Section 3.2: Be careful in your use and discussion of the SAM, if retained.
o It's not clear which behaviour you are saying 'projects onto the positive phase of the SAM'; the wavenumber 3 behaviour or the zonally symmetric annular structure. The Marshall index describes primarily zonally symmetric behaviour, whereas EOF based Antarctic Oscillation indices do capture the 3 centres of action.

o I'm not sure it's useful; in some months (September and November) the circulation does have three centres of action in the expected places. However in October in particular it is not 'SAM-like' at all (either in the zonal-mean or zonally asymmetric sense). I would therefore suggest writing the discussion without reference to the SAM, and concluding the section with a short section, perhaps referencing a map of the SAM's typical behaviour (e.g. Spring SLP regressed onto the Marshall index, as a supplementary figure) and noting the months in which the circulation looked very SAM-like and the remarkable values of the index in this month. It would also be worth checking whether the SAM rankings are broadly robust to use of a different index.

We agree with the reviewer's comment regarding the two paragraphs above. Based on the aforementioned comments, we have removed most of the SAM results from the revised version of the manuscript. We kept just some information in the discussion part (Page 10, lines 25 - 35).

Section 3.3 and throughout: please ensure you are clear and consistent in your use of the terms 'water vapour', 'water vapour transport', 'total column water vapour',' integrated water vapour' etc and their associated acronyms. They were used inconsistently to the extent of my being unable to understand what was being used at all points

We fully agree with this comment. We have changes the terminology to be more clear and consistent.

Section 3.4: To me, the conclusion to be drawn from the maps and indices here (figs 5 and 6) is that there is a lagged local effect between temperature/water vapour and SIC in the regions analyzed. This is perhaps not very surprising although I'm not aware of previous lagged analysis. Some questions to address to interpret the results:

o To what extent is this a manifestation of persistence in SIE anomalies?

Persistence plays definitely a role, but we strongly believe that without a strong atmospheric/oceanic forcing an event like 2016 is rather impossible to occur.

o What are the results for the Weddell Sea in December (where anomalies are greater than in the Ross Sea)? 2016 saw rapid development of WS anomalies in November, implying that at least in this year, it was not just anomaly persistence.

In the revised version of the manuscript we have added a more detailed discussion regarding the Weddell Sea and removed the discussion regarding the Ross Sea, to be consistent in our analysis, especially in the Sections 3.4 and 3.5.

o Does IWV add more (statistical) information, or more physical understanding, over T alone? Figures 5 and 6 show the same regions in the stability maps for IWV and for T, and the timeseries look very highly correlated. A physical discussion and some supporting evidence is needed: is the same circulation bringing in heat and moisture? Or are water vapour anomalies radiatively driving temperature anomalies? Since this seems to me to be the main result of the paper, it is necessary to argue either that the water vapour has some independence from and therefore added predictive power over temperature, or that it adds physical understanding to the sea ice anomalies which cannot be inferred from temperature alone.

Although we strongly agree with this comment, the aim of our manuscript is not to figure out which one of the two variables has a more important role in driving the sea ice anomalies. We strongly feel that such an analysis requires not only observational data, but also model simulations and sensitivity studies. Our aim was to show that extreme warm and moist intrusions are able to trigger negative sea ice anomalies like the one in 2016. We are hoping that our study can be use as an indicator that moisture transport should be considered in future studies regarding the Antarctic sea ice variability. But in order to analyze the contribution of each variable to the sea ice variability, one needs to make designated studies, which is not the aim of our current paper.

o You discuss a dipole of stable regions. This could be related to the ASL, which would cause co-variability in the Ross and Amundsen-Bellingshausen sea, although the footprint in the Weddell sea is larger and further to the east than I'd expect in this case.

ASL could play a role, at least partially, but due to the location of the centers of the dipole we feel that a general discussion is more appropriate compared to relating this just the ASL

variability. As already commented by the reviewer, the footprint in the Weddell Sea is rather different compared to the one observed for ASL.

P7L5: note that the highest SIE over this area is broadly true even when normalized anomalies are used (BAMS state of the climate figure 6.9); i.e. even accounting for natural variability these regions are exceptional in 2016

P7 L7; give the main results from the maps before discussing the indices.

Modified the order as suggested.

Section 3.5:
o It is unclear which water vapour variables are used. Which figures show IWV and which show IWVT? Please check units, acronyms etc.

The text has been modified following these suggestions to make it more clear and easy to follow. All the units and acronyms will be carefully checked.

o I found this hard to follow. I suggest rewriting this section such that each 2-day regional mini case study, is addressed with a little more care (maybe use 'first the ABS',' second the IO',' thirdly the RS'). Can you link these transport events to e.g. cyclones?

In the revised version of the manuscript we have rephrased the aforementioned section to make it more clear and easy to follow.

o Take care over extrapolating to 'decline in first two weeks of December'. The anomalous decline in early December was in the WS (Fig S2) so I don't think you can robustly link it to the event shown.

The text has been modified following the reviewer's suggestions**.**

Section 3.6: This analysis does not add anything to the discussion about 2016, nor much to understanding of SAM-sea ice relationships in general. Table 2 is valuable but I think this discussion could be removed and replaced with a few sentences in the discussion e.g. 'Given the dramatic SAM anomalies in some months, we investigated the long term relationships between regional SIE and the monthly SAM. However, significant relationships were found for only three month-region combinations (of 20) and thus consistent with previous studies […], we find the SAM's role in sea ice variability is complex.' The moving window method could be used to enhance this if the analysis is felt to be critical to the paper. Unless the relationship of the SAM to moisture transport is explicitly addressed, I do not think it adds to the novel scientific content of the paper.
P9 L14; Is this lowest for November, or lowest overall? Is it true for other SAM indices?

As stated above, the discussion regarding SAM has been shortened and changed following the suggestions made by the reviewer. Section 3.6 has been completely removed. Just a small discussion regarding the influence of SAM on the regional SIE variability has been retained in Section 4.

P9 L31; you say you've shown moisture and temperature anomalies could 'also' have led to different anomalies. 'Also' implies it's something different; are the anomalies you've shown

not manifestations of the circulation anomalies discussed? 'poleward advection of warm': you don't show heat transport so this is a slight assumption, although I think Schlosser et al do- please cite.

At the submission date of our paper to ESD (17.11.2017) the Schlosser et al paper (under discussion at that time) does not show figures regarding the heat transport. Nevertheless, the final version of Schlosser et al paper has been recently published, with the addition of the heat transport figure, and now the paper has been cited and referred accordingly. Moreover, in the revised version of the manuscript we have included a new figure representing the vertical integral of northward heat flux and the vertically integrated heat transport from September until December 2016 (Figure 4). This underlines our statements.

P10 L13 ; 'poleward advection of moist and warm air': increased moisture is not necessarily due to increased moisture advection?

We agree with the reviewer's comment and we have modified the text accordingly.

P10 L21; The study of Woods et al was about the Arctic. It's not clear it is relevant here.

The reference has been removed.

**Technical Corrections**

Abstract, Line 13: 'lowest daily sea ice concentration anomalies' -> 'largest magnitude negative daily sea ice concentration anomalies'
Although it's almost certain readers know what SIE is, give the abbreviation at first mention (Pg 1 L25) or at first use in methods section (Pg 2 L23) [rather than at P2 L8 as done in this version].
Page 1 Line 23: Artic -> Arctic
P2, L7: Colins -> Collins
P2, L11: clarify from offset what months 'Austral spring' is.
P3, L20: on time-> in time
P3, L21: Bracketed 'e.g.s' unnecessary and make text more confusing.
P3, L31: Ionita et al (2017) reference should be Ionita (2017)?
P4 : there is inconsistency between use of abbreviations ('WS') and full names ('weddell sea')
P4 L4: 'problematic' a little emotive. Try 'challenging' or 'confusing'?
P4 L5: '8'-> 'eight'
P4 L6: 'positive SIE anomalies over the whole Antarctic region' -> 'positive pan-Antarctic SIE anomalies'. I don't like the phrase 'pan-Antarctic' much, but 'whole' implies 'everywhere'.
P4 L20-21: rewrite e.g. 'In December the whole RS, and most of the ABS and IO were characterised by negative SIC anomalies'
Pg 5 line 5: 'to the zonal' -> 'of zonal'.
P5 L13: 'were'-> 'where'
P5 L19: 'warming' -> 'warm anomalies'
P6 L5: 'particular' -> 'exceptional'
P6 L7: delete sentence 'The rank maps are computed…'
P6 L10: 'to be able to clearly capture…2016 was' -> 'for clarity'
P6 L13: delete 'Figures 3a,b and c indicate that'
P6 L19, L23, L28: 'southern', 'eastern' and 'north-west' should be 'northern', 'western' and 'north-east'? Please check!

P7L6, 16; delete 'respectively'
P7 L9: 'considered'->'consider'
P7 L24; 'December TT'-> 'December TT2'.
P7 L27; delete 'a' before 'predictive'
P8 L2 'there'->'three'
P8 L3; 'employing' -> 'spatially averaging'?
P8 L4 'region'->'regions'
P8 L5: delete comma after 'IWV'
P8 L10: reference Fig 7d
P8 L13: 'BAS'->'ABS'.
P8 L24: 'trimming'->'timing'.
P8 L17: Reference Figure S2.
P9 L15 'led to'->'associated with' (the SAM is in part expressing the behaviour of the westerlies. It is not a physical driver.).
P9 L31 'lead'->'led'.
P10 L16 'this'->'these'
P10 L16 'moist'->'moisture'.
P10 L21; 'In recent' ->'In a recent'.

All the aforementioned technical corrections have been addressed and inserted in the revised version of the manuscript. We are thankful for the very careful reading of the reviewer at this point.

Corrections: Figures

Figure 1: Optional, but inclusion of topography and lines of latitude could make this a more useful reference figure for readers not as familiar with the Antarctic.

A figure with topography does not add much value to our overall Figure 1, mainly because we are focusing our analysis on the sea ice anomalies.

Figure 2: The vectors are too small to be legible, and are not described in the caption. Repeating sector names in-plot is unnecessary here and the legibility is poor, so I suggest removing.

The figure has been changed according to the suggestions. We have removed the sector names from Figure 2.

Figure 3: As Figure 2 for sector labels. Remove '2016' in subfigure label titles. I also suggest you change the caption text to 'Ranking of 2016 monthly: sea ice concentration (SIC, first column, '1' means lowest SIC in period); integrated water vapour (IWV, second column, '1' means wettest month in period); temperature (TT, third column, '1' means warmest month in period). The period is 1979-2016. Rankings below 5 appear white."

The aforementioned suggestions have been taken into account and the figure caption for Figure 3 (now Figure 5) has been modified accordingly.

o Technical question; how do repeated '0's rank (i.e. if several years have SIC=0?)

The years with SIC = 0 are treated as NaN's and they are not taken into account.

o The colour scale is probably not colour-blind friendly. Please improve. Figure 5 and 6

We have tried to find a color-blind color scheme (see new figure 5 and S3).

o The correlations in e) and f) would be clearer if the SIE anomaly were negated.

We have negated the SIE anomalies, but the new figure is more difficult to follow compared to the original one, as such we have decided to keep the original figure.

o Add title to color bar.

Done.

o Simplify caption text for panels a-d to (Fig 5 example): 'The stability maps between November Indian Ocean (IO) sea ice extent and a) October IWV, b) November IWV, c) October TT, d) November TT.

Modified as suggested.

o Please check panel refs in caption to Fig 6; both time series panels referred to as c).

Done.

Figure 7

o Check journal standards on color scale (avoiding use of green-red for example).
o It would be helpful to show again in this figure the squares used to define regions.
o What are the vectors?
o Please clarify the dates shown in time series (are all days shown or only October and November) and check use and consistency of IWT/IWVT etc (see my general comments) and units.

The figure has been modified following the reviewer's suggestions. The only thing that we did not take into account was to add the delimiting lines for the sea ice extent regions. Adding this line will make the figure almost impossible to follow.

Supplementary Figure 1: In caption, mention 2016 just once, and state what the red number in each panel is (the ranking of that month's regional SIE).

Modified as suggested.

SF2: It would be helpful to include full regional names once more and to either change colors or find another way of making them clearer (IO, ANT and WS are very hard to distinguish) and of separating the full Antarctic time series from the individual regions. Y axis: Sie-> SIE.

Modified as suggested.

SF3: Interpolation of contours here is unhelpful as there is no propagation of anomalies in the x-direction. I suggest using a block rather than interpolated contour. Also please change colour scale and use less classes for legibility.

Figure S2 has been removed from the revised version of the manuscript.

SF4: See comments on (main text) Figure 3 caption and color scale.

Modified as suggested.

Response Reviewer 2

We would like to thank the reviewer for the positive feedback on our manuscript and we are grateful for the comments on how it can be further improved. We provide below a point by point response to the reviewer comments/suggestions.

**Summary and overall comments**
The manuscript describes the dramatic retreat of Antarctic sea ice in Oct-Dec 2016 in detail, and relates it to anomalies in the atmospheric circulation, in particular surface temperature, total column water vapor and total column water vapor flux. The methodology was previously unknown to me, and the results are interesting, warranting publication in ESD. However, the writing in the manuscript needs improvement. I therefore recommend publication after minor revisions.

**Specific comments**
Fig. 2: Please indicate in the caption what is shown. What are the arrows in left column plots? In right column, are colours IMV (integrated water vapour) and arrows the IMVT (integrated water vapour transport)? Also, please insert reference arrow.

Figure 2 caption has been modified following the reviewer's suggestions.

Fig 3: For comparison with Fig 2, please swap center and right columns so that temperature is in middle and IMV to the right.

We have modified Figure 2 and added two new figures (Figure 3 and Figure 4) so Figure 2 and 3 are distinct and cannot be compared directly in the revised version of the manuscript.

Fig 4: This figure is referenced on lines 26, 29 on page 3, but I think you mean Fig 5 or 6? In that case, Fig 4 is only referenced to once in the paper, and is somewhat unnecessary. The relation between high temperatures and low sea ice is already made in Fig 3. I strongly suggest to remove it or add it to Supp. material.

We decided to keep figure 4 (now Figure 6) mostly because is based on observation data, thus giving more confidence regarding the reanalysis data.

Fig. 6: From Fig 6a,c, my guess is that there is an SLP anomaly somewhere over the Antarctic Peninsula driving the changes in advection. Hence, the 2nd box is correlated to WS sea ice, but is likely not causing the anomalies in WS sea ice? Also, the caption describes Fig 6c three times, but never Fig 6e or 6f.

The text and figure captions have been modified following the reviewer's suggestions.

Page 1, Line 11,12: Throughout the paper, the authors use the unusual abbreviations "IWV" for integrated water vapour and "TT" for 2m temperature. For the former, "VIWV" (vertically integrated water vapour) would be a better name so that the direction of integrations is explicit, or the authors could use "TCWV" (total column water vapour) or "IWVC" (integrated

water vapour column) which are more common. Rather than "TT", perhaps "T2m" or "Ts" would be better?

We strongly agree with this issue, and in the revised version of the manuscript we have improved the description of each variable and their corresponding abbreviation to make the paper easier to read. Each variable we use has been properly abbreviated in the data section. The text has been checked for inconsistencies and modified accordingly.

Page 1, Line 32: The sentence "Antarctica experienced..." could be changed to "Between September 2016 and April 2017, Antarctica experienced a dramatic shift featuring strong negative SIE anomalies".

The text has been modified accordingly.

Line 2, Line 2: "could be caused either by a potential change..". What is meant by "potential change"? Perhaps re-phrase to "could potentially have been caused by a change in the long-term trend or by natural climate variability"?

The text has been rephrased following the reviewer's suggestion.

Page 2, Line 20: Skip the comma between "utilize" and "the".

Done.

Page 3, Line 20: "does not change on time"? Do you mean "is time-independent"?

The text has been rephrased following the reviewer's suggestion.

Section 2.2: The method is interesting, but the description may need some clarification. If there are 37 years of data and you use a 21-year moving window, then you only have 37-21=16 data points in time? Does this mean that the maps in Fig. 5,6 are made from 16 time levels? And 80% of 16 is about 13, so you require significant correlations for 13 of 16 time levels?

A justification regarding the use of stability maps and their robustness with short time series has been added in the revised version of the manuscript (see Section 2.2 Methods).

Page 3, Line 26, 29: Should be Figure 5,6 instead of 4,5?

Yes indeed. The text has been modified accordingly.

Page 4, Line 22: Suggest "striking" instead of "very particular"

Modified as suggested.

Page 4, Line 29, 30: Perhaps use "SIE anomaly below −2.2 mil. km2" instead of "more than 2.2" so that it is clear that the anomalies are negative. Same at Line 30.

Modified as suggested.

Page 5, Line 2: "1980's"

Modified as suggested.

Page 5, Line 12: At several places in the paper (e.g. Page 6, Line 32 and Page 7, Line 12), the authors use "positive temperature anomalies and enhanced water vapour". Why not use "positive anomalies in surface temperature and total column water vapour"?

In the revised version of the manuscript the text has been modified following the reviewer's suggestion.

Page 6, Lines 9, 15, 19, 20, 23: "wet" often refers to high precipitation. Use "most humid" instead? Also, do we know if these events are linked to high values of precipitation?

Throughout the revised version of the manuscript we use now "moist" instead of "wet". We agree that wet usually is associated with precipitation, thus the text has been modified accordingly.

Page 6, Line 32: Suggest change from "extreme warm temperatures and enhanced moisture" to "large positive anomalies in surface temperature and total column water vapour".

Modified as suggested.

Page 8, Lines 1-5: I found these sentences unclear. I would suggest "For October and November each year, we calculate daily IWVT (or TCWV or IWVC, see above comment) over three areas in the ABS, IO, RS respectively. The areas are those which rank as the warmest and most humid on record in 2016, and are marked in Figures 3e and 3h. The time series of IWVT for these areas are shown in Figures 7a, c, e. "

In the revised version of the manuscript we have modified almost entirely section 3.5 for a better reading.

Page 8, Line 8: "reaching the" rather than "penetrating until".

Modified as suggested.

Page 8, Line 13: "ABS" instead of "BAS"?

Modified as suggested.

[revised manuscript text omitted]